# Risk correlation identification of futures market based on wavelet transform and quantile Granger causality test

Zi Qian Wu[ORCID]*

Tibet University, Lhasa, China

* qian1962388262@sina.com

## Abstract

Futures market is an important part of the financial market, with a high degree of liquidity and leverage effect. However, the futures market is also faced with various risk factors, such as price fluctuations, market shocks, supply and demand changes. In order to better determine the risk correlation between specific futures markets, this paper uses the wavelet transform—quantile Granger causality test method to identify the risk correlation of four major futures markets in the US futures market from the end of January 2009 to the end of March 2023, such as gold, crude oil, soybeans and natural gas. It provides a new perspective and method for the risk correlation identification of the futures market. The results show that futures contracts with different maturities and price fluctuations under different quantiles have a significant impact on risk correlation. Specifically, in 1-month and 6-month futures contracts, the strongest bidirectional risk correlation exists between gold and natural gas (T-statistics -15.94 and 10.92, respectively); In the 1-month futures contract, there is also a strong bidirectional risk association between crude oil and soybeans and natural gas (T-statistics are 6.87, 17.42, -2.05, 7.35, respectively), while in the 6-month futures contract, there is a bidirectional risk association between crude oil and soybeans (T-statistics are -2.49 and 18.374, respectively). However, natural gas has unidirectional risk association with crude oil and soybean (t statistics are 2.7 and -3.35, respectively); There is a bidirectional risk correlation between gold and soybean, that is, the risk correlation between gold and soybean increases with the increase of the degree of price fluctuation; There is a one-way risk association between gold and crude oil, soybean and gold, and crude oil and natural gas (the T-statistic is greater than the critical value of 1.96). In addition, there is a strong bidirectional or unidirectional risk association between all varieties at the 0.95 quantile. The research results of this paper have certain reference value for the supervision, investment and risk management of the futures market. This paper uses the wavelet transform and quantile Granger causality test method to identify the risk correlation of the futures market, providing a new perspective and method for the risk correlation identification of the futures market, and uses relatively new data to ensure the effectiveness of the empirical analysis. However, there are some limitations in this paper, such as the applicability of wavelet transform-quantile Granger causality test method. Future studies can further expand the sample range,

**Data Availability Statement:** I've put the data uploaded to figshare, specific DOL is as follows: 10.6084/m9figshare.24085971.

**Funding:** The author(s) received no specific funding for this work.

**Competing interests:** The authors have declared that no competing interests exist.

compare the effects of different methods, and explore the risk transmission mechanism between different varieties.

## 1 Introduction

Is there a risk correlation between different varieties in the futures market? If so, how does the risk correlation change with different frequency bands and different quantiles? How can risk correlations in futures markets be effectively identified and measured? This is the research question that this paper tries to answer. The futures market is an important part of the financial market. It not only provides price discovery, hedging and risk transfer functions for producers, consumers and investors, but also plays an important role in the development and stability of national economy. However, there are also a lot of uncertainties and risks in the futures market, especially in the context of globalization and financialization, price fluctuations between different varieties may have mutual influence and contagion effect, resulting in increased volatility of the futures market and intensified systemic risks. Therefore, identifying and analyzing the risk correlation between different varieties in the futures market has important theoretical significance and practical value for revealing the operating mechanism of the futures market, optimizing the regulatory policy of the futures market, and improving the investment efficiency and risk management level of the futures market.

The purpose of this paper is to use a novel and effective method—wavelet transform and quantile Granger causality test method to identify and measure the risk correlation of four main varieties (gold, crude oil, soybean and natural gas) in China's futures market, and reveal their differences and changes under different frequency bands and different quantiles.

The method of wavelet transform and quantile Granger causality test is a combination of wavelet transform and quantile Granger causality test, which has the following advantages: (1) Wavelet transform can decompose time series into different levels of approximate components and detailed components, so that it can be analyzed in time domain and frequency domain at the same time to capture the characteristics of time series in different scales; (2) quantile Granger causality test can be tested on different quantiles, so that the characteristics of time series under different risk levels can be considered; (3) The wavelet transform quantile Granger causality test can overcome the dependence of traditional methods on the assumptions of linearity, stationarity, homogeneity, symmetry, etc., and is more suitable for processing complex and changeable financial data.

In order to better determine the risk correlation between specific futures markets, the main contributions of this paper are as follows: (1) This paper uses the method of wavelet transform-quantile Granger causality test to identify the risk correlation of futures markets, providing a new perspective and method for the risk correlation identification of futures markets; (2) This paper not only analyzes the existence and characteristics of risk correlation in futures market as a whole, but also analyzes the differences and changes of risk correlation in futures market from different frequencies and different risk levels, providing more detailed and comprehensive information for the supervision, investment and risk management of futures market; (3) This paper uses relatively new data to ensure the validity of the empirical analysis.

## 2 Literature review

Domestic and foreign scholars have carried out a lot of discussion and analysis on the research of risk correlation identification in futures market, mainly using the following methods:

Cointegration analysis: Cointegration analysis is a method used to test whether there is a long-term equilibrium relationship between time series, it can avoid the problem of pseudo-regression, and can take into account the characteristics of non-stationarity and nonlinear. Representative studies of cointegration analysis include Engle and Granger [1], Johansen [2], Johansen and Juselius (1990) [3], etc. In terms of risk correlation identification in futures market, cointegration analysis is mainly used to test whether there is a long-term equilibrium relationship between different varieties, as well as the stability and strength of the equilibrium relationship. For example, Chen et al. [4] used cointegration analysis and error correction model to test the price co-integration relationship among soybean, corn, cotton and sugar in China's futures market, and found that there is a stable long-term equilibrium relationship, and the price of soybean has a significant impact on the prices of the other three agricultural products. Wang et al. [5] used cointegration analysis and vector error correction model to test the price co-integration relationship among copper, aluminum and zinc non-ferrous metals in China's futures market, and found that there were two stable long-term equilibrium relationships, and copper prices had a significant impact on the prices of other two non-ferrous metals. Yao Haixiang et al. [6] use the cointegration test and Granger causality test to analyze the impact of night trading on China's agricultural futures market, and find that night trading has a significant enhancement effect on China's agricultural futures market, and there is a two-way causal relationship between night trading and day trading. This method can consider the long-term equilibrium relationship and short-term causality between the two variables, but it also needs to determine the appropriate cointegration order, hysteresis order and stability test.

Vector autoregressive model: A vector autoregressive model is a method used to describe the dynamic interaction between multiple time series, which can simultaneously consider the hysteresis effect of each time series on itself and other time series, while using tools such as impulse response function and variance decomposition to analyze issues such as shock propagation effect and contribution degree. Representative studies of vectorial autoregressive models include Sims (1980) [7], Lutkepohl [8], Hamilton (1994) [9], etc. In terms of risk correlation identification in futures market, vector autoregressive model is mainly used to test whether there is a short-term dynamic relationship between different varieties, as well as the direction and intensity of the dynamic relationship. For example, Zhang et al. [10] used the vector autoregressive model and impulse response function to test the dynamic price relationship among soybean, corn, cotton and white sugar in China's futures market, and found that soybean prices had a significant positive impact on the prices of the other three agricultural products. While the price of corn has a significant negative impact on the prices of the other three agricultural products; Li et al. [11] used vector autoregressive model and variance decomposition to test the dynamic price relationship between copper, aluminum and zinc in China's futures market, and found that copper price contributed the most to the price fluctuation of the other two non-ferrous metals, while zinc price contributed the least to the price fluctuation of the other two non-ferrous metals. Liu Jianghao et al. [12] used VAR model to analyze the impact effect of RMB exchange rate on domestic soybean price, and found that RMB exchange rate fluctuation had a significant impact on domestic soybean price, and the impact lasted for a long time. This method can consider the interaction and dynamic effect between multiple variables, but it also needs to determine the appropriate variable selection, lag order, stability test, etc.

Covariance matrix decomposition: Covariance matrix decomposition is a method used to extract the characteristics of common fluctuations between multiple time series, which can decompose the covariance matrix into eigenvalues and eigenvectors, so that information such as principal components and factors can be obtained. Representative studies of covariance matrix decomposition include Hotelling [13], Anderson (1984) [14], Jolliffe [15], etc. In the

risk correlation identification of futures market, covariance matrix decomposition is mainly used to test whether there are common volatility characteristics among different varieties, as well as the source and influence of common volatility characteristics. For example, Liu et al. [16] used covariance matrix decomposition and principal component analysis to test the volatility characteristics among ten varieties (including agricultural products, non-ferrous metals, energy and chemical industries, etc.) in China's futures market, and found that there is a principal component that can explain most of the volatility changes, and this principal component has a strong correlation with international market factors. Wang et al. [17] used covariance matrix decomposition and factor analysis to test the volatility characteristics among 11 varieties (including agricultural products, non-ferrous metals, energy and chemical industries, etc.) in China's futures market, and found that three factors could explain most of the volatility changes. And these three factors are strongly correlated with domestic macroeconomic factors, international market factors and industry characteristics factors, respectively.

Dynamic conditional correlation model: The dynamic conditional correlation model is a method used to describe the correlation between multiple time series over time. It can decompose the covariance matrix into two parts: conditional variance and conditional correlation coefficient, so that the conditional correlation coefficient can be modeled and estimated. Representative studies of dynamic conditional correlation models include Engle [18], Tse and Tsui [19], Aielli [20], etc. In terms of risk correlation identification in the futures market, the dynamic conditional correlation model is mainly used to test whether there is a correlation between different varieties over time, as well as the change rule of correlation and influencing factors. For example, Chen et al. [21] used the dynamic conditional correlation model and impulse response function to test the dynamic correlation among 12 varieties (including agricultural products, non-ferrous metals, energy and chemical industries, etc.) in China's futures market, and found that the correlation between different varieties showed different correlations in different time periods. Steel [22] introduced the application of model averaging method in economics, pointing out that model averaging method can effectively deal with model uncertainty and parameter uncertainty, and improve the accuracy and robustness of prediction. This method can synthesize the information of multiple models, but it also needs to determine the appropriate model set, model weight, model selection criteria, etc. Cai [23] used dynamic model averaging method to establish a volatility prediction model based on dynamic quantile regression. Xiong Tao and Bao Yukun [24] also used dynamic model averaging method to predict soybean futures price, and found that this method can effectively improve the accuracy and robustness of soybean futures price prediction.

Other methods of market risk identification mainly include: differential, CoVaR, GARCH family, quantile regression, event study, etc. For example, He Chaofei and Yu Leng [25] used the differential method to evaluate the impact of changing the temporary purchase and storage policy to target price system on China's soybean planting area; Ke et al. [26] used CoVaR method to measure the risk transmission between the agricultural futures markets of China and the United States, and found that there was two-way risk transmission in the agricultural futures markets of China and the United States. Both Liu Wenchao [27] and Meng Bin [28] used GARCH family to describe the risk spillover; Hau et al. [29]. used quantile-to-quantile regression method to analyze the impact of crude oil price fluctuations on China's agricultural futures market, and found that crude oil price fluctuations had a heterogeneous dependence on China's agricultural futures market. Liu Liyu [30] used the event study method to analyze the volatility and impact of international soybean prices under the Sino-US trade war, and found that the Sino-US trade war led to a significant decline in international soybean prices.

From the perspective of content: Some literature focuses on the risk spillover between domestic crude oil futures and other financial assets. For example, Ma Yanran et al. [31], Liu

 

Yinglin et al. [32], Tian Hongzhi et al. [33], Zhang Dayong et al. [34], WU B B [35], Meng Juan et al. [36], Ahmed A D et al. [37] and Li Zhihui et al. [38]; Some literature focuses on the risk spillover between the international crude oil market and Chinese commodity futures, such as YANG Y Y et al. [39], Lv Fang et al. [40] and Jiang Yun et al. [41].

However, the methods in the above literatures also have some limitations, such as ignoring features such as nonlinearity and non-stationarity, assuming conditions such as homogeneity and symmetry, and lacking differentiation of different frequencies and risk levels. Therefore, this paper attempts to introduce a novel and effective method, wavelet transform and quantile Granger causality test, to identify the risk correlation of the return series of four major varieties (gold, crude oil, soybean and natural gas) in China's futures market. This paper believes that this is a meaningful and challenging research gap, which is worthy of further exploration.

The main structure of this paper is as follows: The first part is introduction and literature review; The second part is the model setting, the third part is the empirical analysis; The fourth part is the conclusion and policy recommendations.

## 3 Model setting

In order to better identify the risk correlation between futures markets, the risk correlation identification model of wavelet transform and quantile Granger causality test established in this paper mainly includes two parts, namely the wavelet transform part and the quantile Granger causality test part.

### 3.1 Wavelet transform

First, for a time series $y_{et}$, the wavelet basis function $\psi$ and the number of decomposition layers J are used for discrete wavelet transformation, and the approximate coefficients $a_J$ and detail coefficients $d_J$ can be obtained, satisfying the following relations:

$$y_t = \sum_{k=-\infty}^{\infty} c_{j,k} \phi_{J,K}(t) + \sum_{j=1}^{J} \sum_{k=-\infty}^{\infty} d_{j,k} \psi_{j,k}(t) \tag{1}$$

$$\psi_{J,K}(t) = 2^{-j/2} \psi(2^{-jt} - k) \tag{2}$$

Where $\phi_{J,K}(t)$ and $\psi_{j,k}(t)$ are scale functions and wavelet functions generated from the wavelet basis function $\psi$.

$c_{j,k}$ and $d_{j,k}$ are the approximate coefficients and the detail coefficients, which can be obtained by the following formula:

$$c_{j,k} = \int y_t \phi_{J,K}(t) dt \, d_{j,k} = \int y_t \psi_{j,k}(t) dt \tag{3}$$

$c_{0,k}$ represents the approximation coefficients at the lowest frequency and the largest scale, and $d_{j,k}$ represents the detail coefficients at the highest frequency and the smallest scale.

The approximate and detail components can be obtained by the following formula:

$$A_j(t) = \sum_{k=-\infty}^{\infty} c_{j,k} \phi_{J,K}(t) \, D_j(t) = \sum_{k=-\infty}^{\infty} d_{j,k} \psi_{j,k}(t) \tag{4}$$

Where, $A_0(t) = A_j(t)$, $D_0(t) = 0$

$A_j(t)$ represents the approximate component of the j layer, and $D_j(t)$ represents the detailed component of the j layer.

## 3.2 Quantile Granger causality test

Quantile Granger causality test established in this paper is mainly divided into the following steps:

First, the lag variable matrix is constructed, that is, the value of the past lag period of the two time series is taken as the independent variable.

Second, quantile regression of the constrained model is carried out, which only includes the lag variable of y, and the regression coefficient and residual are obtained.

Thirdly, quantile regression of the unrestricted model is carried out, including the lag variables of y and x, to get the regression coefficient and residual.

Fourth, calculate the sum of squares of the residuals of the restricted and unrestricted models.

Fifth, calculate the T-test statistic, which is the difference of the sum of squares of the residuals of the two models divided by lag and then divided by the sum of squares of the residuals of the unrestricted model divided by the degrees of freedom.

Sixth, compare the T-value, which is used to describe the degree of risk association between different futures varieties.

Suppose $y_t$ and $x_t$ are two time series, q is quantile level, k is lag order, $\beta_q$ and $\gamma_q$ are quantile regression coefficient vectors, $\in_t$ and $\eta_t$ are quantile regression residual vectors, then

$$y_t = \beta_q^T y_{t-1} + \epsilon_t (Restricted\ model) \tag{5}$$

$$y_t = \gamma_q^T (y_{t-1}, x_{t-1}) + \eta_t (Unrestricted\ model) \tag{6}$$

Where $y_{t-1} = (y_{t-1}, y_{t-2}, \cdots y_{t-k})^T, , .x_{t-1} = (x_{t-1} x_{t-2}, \cdots x_{t-k})^T$

Quantile regression coefficient vectors $\beta_q$ and $\gamma_q$ can be obtained by minimizing the following objective function:

$$\min_{\beta_q} \sum_{t=k+1}^{n} \rho_q (y_t - \beta_q^T y_{t-1}) \tag{7}$$

$$\min_{\gamma_q} \sum_{t=k+1}^{n} \rho_q (y_t - \gamma_q^T (y_{t-1}, x_{t-1})) \tag{8}$$

$$\rho_q(u) = u(q - I(u < 0)) \tag{9}$$

The quantile regression residual vectors $\in_t$ and $\eta_t \eta\_t$ can be obtained by the following formula:

$$\in_t = y_t - \beta_q^T y_{t-1} \tag{10}$$

$$\eta_t = y_t - \gamma_q^T (y_{t-1}, x_{x-1}) \tag{11}$$

The sum of the squares of the residuals of the restricted and unrestricted models can be obtained by the following formula:

$$RSS_R = \sum_{t=k+1}^{n} \epsilon_t^2 \tag{12}$$

$$RSS_{UR} = \sum_{t=k+1}^{n} \eta_t^2 \tag{13}$$

The test statistic can be obtained by the following formula:

For two time series and $x_t$, and a quantile level $q \in (0,1)$, quantile Granger causality test is performed, and the following t-test statistic $t_q$ can be obtained:

$$t_q = \frac{(RSS_r - RSS_{ur})/L}{RSS_{ur}/(T - 2L)} \tag{14}$$

Where, $RSS_R$ and $RSS_{UR}$ are the sum of squares of residuals of restricted and unrestricted models respectively, L is the order of lag, T is the sample length, t(L, T−2L; $t_q$) is the value of the cumulative distribution function of the T distribution with L and T−2L degrees of freedom at point $t_q$. If the value of the t statistic is greater than the critical value corresponding to the 5% significance level, such as 1.96, then we can reject the null hypothesis and think that x has quantile-dependent Granger causality on y, that is, there is a quantile-dependent risk association.

## 4 Empirical analysis

### 4.1 Data selection and explanation

Since this paper builds a model based on wavelet transform and quantile Granger causality test to measure the risk correlation of futures markets, in order to better explain the risk correlation among American futures markets, this paper mainly selects the American precious metal futures market, crude oil futures market, grain futures market and common energy futures market. Therefore, this paper selects the representative commodities of these major futures markets, mainly including: gold futures market price (gold), Brent crude oil futures market price (oil), soybean futures market price (Syean) and natural gas futures market price (NG). The futures contract term is 1 month term and 6 month term, and the time interval is as follows: The data from the end of January 2009 to the end of March 2023 are all derived from Bloomberg database.

From Table 1, we can see that the average price of gold futures is the highest, at $1,427.25 / oz, and also the most volatile, with a standard deviation of $276.481 / oz. This shows that the price of gold futures is affected by many factors, such as market demand, inflation, geopolitics, the exchange rate of the US dollar and so on. The lowest price for gold futures was $887.1 an ounce in January 2009, when the world was reeling from the financial crisis. The highest price for gold futures was $1,994 in August 2022, when the world was facing the fourth wave of the coronavirus outbreak, leading to market panic and increased demand for safe havens.

The average price of crude oil futures is $70.90 / barrel, which is highly volatile with a standard deviation of $22.014 / barrel. This shows that the price of crude oil futures is affected by supply and demand, the policies of oil-producing countries, and environmental factors. The lowest price for crude oil futures was $19.09 per barrel in April 2020, when global oil demand plummeted due to the COVID-19 pandemic and a price war between oil producing countries caused a glut in the market. The highest price for crude oil futures was $115.26 in April 2011, when political unrest in the Middle East and North Africa raised fears of oil supply disruptions.

**Table 1. Descriptive statistics of 1-month futures.**

| VarName | Mean | SD | Min | Max |
|---|---:|---:|---:|---:|
| Gold futures | 1427.25 | 276.481 | 887.1 | 1994 |
| Crude oil futures | 70.90 | 22.014 | 19.09 | 115.26 |
| Soybean futures | 1148.04 | 242.224 | 840.75 | 1753 |
| Natural gas futures | 3.54 | 1.257 | 1.643 | 9.141 |

The average price of soybean futures is $1,148.04 / ton, which is less volatile and has a standard deviation of $242.224 / ton. This shows that the price of soybean futures is affected by weather, harvest, inventory, trade policy and other factors. The lowest price for soybean futures was $840.75 a ton in February 2009, when the global economic slowdown reduced demand for soybeans. The highest price for soybean futures was $1,753 a ton in September 2012, when the United States was hit by a severe drought that affected soybean production and quality.

Natural gas futures averaged $3.54 / mmBTU, with the least volatility and a standard deviation of $1.257 / mmBTU. This shows that the price of natural gas futures is affected by seasonal demand, storage levels, alternative energy sources and other factors. The lowest price for natural gas futures was $1.643 per million British thermal units in June 2020, when the coronavirus pandemic reduced demand for industrial and commercial gas. The highest price for natural gas futures was $9.141 in January 2009, when a cold snap in North America boosted demand for residential gas.

From Table 2, we can see: The average price of gold futures is the highest, at $1446.45 / oz, and also the most volatile, with a standard deviation of $284.349 / oz. This shows that the price of gold futures is affected by many factors, such as market demand, inflation, geopolitics, the exchange rate of the US dollar and so on. The lowest price for gold futures was $900.1 an ounce in February 2009, when the world was reeling from the financial crisis. The highest price for gold futures was $2,072 in February 2023, when the world was facing the fifth wave of the coronavirus outbreak, causing market panic and rising safe-haven demand.

The average price of crude oil futures is $71.35 / BBL, which is volatile with a standard deviation of $20.315 / BBL. This shows that the price of crude oil futures is affected by supply and demand, the policies of oil-producing countries, and environmental factors. The lowest price of crude oil futures was $28.72 per barrel in May 2020, when global oil demand fell sharply due to the COVID-19 pandemic and oil producers reached an agreement to cut production, resulting in a shortage of supply. The highest price for crude oil futures was $114.78 in April 2011, when political unrest in the Middle East and North Africa raised fears of supply disruptions.

The average price of soybean futures is $1,097.38 / ton, which is less volatile and has a standard deviation of $176.818 / ton. This shows that the price of soybean futures is affected by weather, harvest, inventory, trade policy and other factors. The lowest price for soybean futures was $851 a ton in March 2009, when the global economic slowdown reduced demand for soybeans. The highest price for soybean futures was $1,506.5 a tonne in August 2012, when the US was hit by a severe drought that affected soybean production and quality.

Natural gas futures averaged $3.72 / mmBTU, with the least volatility and a standard deviation of $1.133 / mmBTU. This shows that the price of natural gas futures is affected by seasonal demand, storage levels, alternative energy sources and other factors. The lowest price for natural gas futures was $1.926 per million British thermal units in July 2020, when the coronavirus pandemic reduced demand for industrial and commercial gas. The highest price for natural gas futures was $8.184 in January 2009, when a cold snap in North America boosted demand for residential gas.

**Table 2. Descriptive statistics of 6-month futures.**

| VarName | Mean | SD | Min | Max |
|---|---|---|---|---|
| Gold futures | 1446.45 | 284.349 | 900.1 | 2072 |
| Crude oil futures | 71.35 | 20.315 | 28.72 | 114.78 |
| Soybean futures | 1097.38 | 176.818 | 851 | 1506.5 |
| Natural gas futures | 3.72 | 1.133 | 1.926 | 8.184 |

**Table 3. US 1-month futures contract unit root test.**

| Variables | ADF test | PP inspection |
|---|---|---|
| Gold Futures | 0.000 * * * | 0.000 * * * |
| Crude oil futures | 0.000 * * * | 0.000 * * * |
| Soybean futures | 0.000 * * * | 0.000 * * * |
| Natural gas futures | 0.000 * * * | 0.000 * * * |

Note: *** indicates passing the 1% significance level test

## 4.2 Unit root test

As can be seen from Tables 3 and 4, the P-values of all four futures contracts (gold, crude oil, soybeans and natural gas) are very close to zero, far less than the significance level of 0.05, regardless of whether the ADF test is used or the PP test. This means that we can reject the null hypothesis of the existence of a unit root and consider all four futures contracts to be stationary time series.

## 4.3 Cointegration test

As can be seen from Tables 5 and 6, there is a certain degree of long-term relationship between variables such as 1-month futures contract and 6-month futures contract, gold futures, crude oil futures, soybean futures, natural gas futures, etc.

## 4.4 Identification of risk associations in the futures market with a one-month term

### 4.4.1 Risk association identification of futures market with 1-month term based on approximate component. From Fig 1 we can see:

Gold and natural gas have a strong direct two-way risk association, which may be due to more optimistic expectations of future economic growth, increased investor demand for various assets, and a weaker US dollar; And at the high risk level, the price changes of gold and natural gas have a strong negative correlation, because gold is regarded as a safe haven asset, natural gas is regarded as a cyclical asset, when the market risk increases, investors will turn to gold and give up natural gas. The prices of these two commodities are also affected by a variety of factors, such as global supply and demand, political economy, and market sentiment, resulting in the correlation between them increasing and weakening at times.

There is a strong direct two-way risk correlation between crude oil and soybeans and natural gas, possibly because crude oil has an impact on other energy commodities and agricultural products, while soybeans have an impact on other oils and fats and renewable energy. The

**Table 4. US 6 month futures contract unit root test.**

| Variables | ADF test | PP inspection |
|---|---|---|
| Gold Futures | 0.000 * * * | 0.000 * * * |
| Crude oil futures | 0.000 * * * | 0.000 * * * |
| Soybean futures | 0.000 * * * | 0.000 * * * |
| Natural gas futures | 0.000 * * * | 0.000 * * * |

Note: *** indicates passing the 1% significance level test

Table 5.  1 month futures contract cointegration test.

| Variables | Whether to pass cointegration |
|---|---|
| gold[1] | Yes |
| Oil[1] | Yes |
| Syean[1] | Yes |
| NG[1] | Yes |

Table 6.  6-month futures contract co-integration test.

| Variables | Whether to pass cointegration |
|---|---|
| gold[6] | Yes |
| oil[6] | Yes |
| Syean[6] | Yes |
| NG[6] | Yes |

interactions between these commodities form a complex web in which they move in tandem or cancel each other out to some extent.

The indirect two-way risk correlation between gold and crude oil may be due to the fact that they are both affected by multiple factors such as the dollar exchange rate, global economic conditions, geopolitical risks and inflation expectations, which sometimes cause them to move in the same direction and sometimes cause them to move in the opposite direction. The interactions between these factors form a complex web that causes them to exhibit changes in positive and negative correlations to a certain extent.

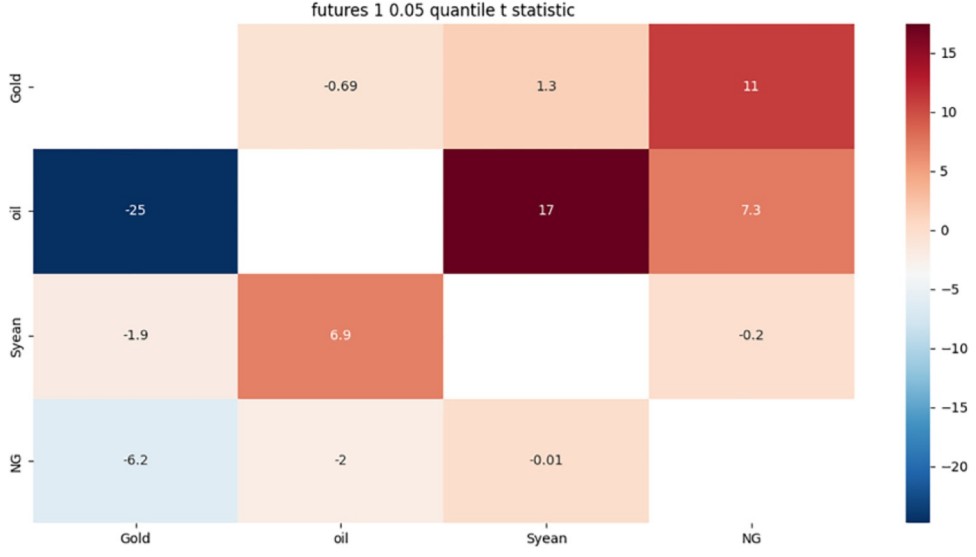

**Fig 1.  Risk correlation heat map of 0.05 quantile for 1 month approximate component.** Note: The blank part in the figure indicates that the risk correlation t statistic value of the futures price itself is 0, Gold represents the gold futures price, oil represents the Brent crude oil futures price, Syean represents the soybean futures price, and NG represents the natural gas futures price. When the price of a certain market has a small fluctuation, the price of other markets will be greatly affected. The larger the t statistic, the stronger the risk correlation, and the smaller the risk correlation, the weaker the risk correlation.

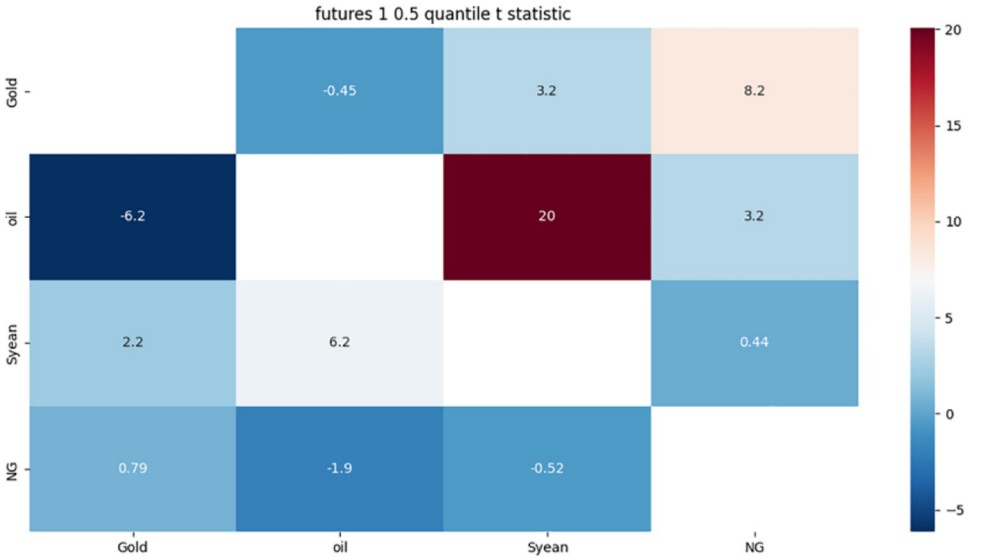

**Fig 2. 1-month approximate component 0.5 quantile risk correlation heat map.**

It can be seen from Fig 2 that gold and soybeans have a significantly weaker bidirectional risk association, mainly because they are both affected by a number of common or mutually affecting factors, including the global economy, politics, monetary policy, US dollar exchange rate, climate change, investor sentiment, etc.

There is a strong bidirectional risk correlation between crude oil and soybean. The reasons mainly include the impact of crude oil price on soybean production cost, biofuel demand, global economic and trade situation, US dollar exchange rate, and the interaction among these factors. These factors will lead to the same direction or reverse changes in crude oil and soybean prices, thereby increasing the risk transmission and amplification between the two.

There is a strong one-way risk correlation between gold and crude oil, mainly because of three aspects: first, changes in the exchange rate of the US dollar have the same impact on the prices of both; second, rising crude oil prices lead to inflation expectations and economic uncertainties, prompting investors to buy gold as a hedge and hedge asset; third, crude oil exporting countries increase the demand for gold to transfer risks and preserve value. These factors make the price of gold more sensitive to the change of crude oil price, while the price of crude oil may not have obvious feedback to the change of gold price.

Natural gas has a strong one-way risk correlation with both gold and crude oil, mainly due to three aspects: first, the change of energy cost has an impact on the production and supply of both; second, the change of energy demand has an impact on the consumption demand of both; third, the change of US dollar exchange rate has an impact on the import, export and price of both. These factors make the price of natural gas more sensitive to the changes of gold and crude oil prices, while the price of gold and crude oil may not have obvious feedback to the changes of natural gas prices.

From Fig 3:

Gold, crude oil and soybeans all have strong bidirectional risk correlation for four main reasons: First, the change of the US dollar exchange rate has the same direction impact on the prices of the three; second, the price of crude oil has an impact on the production, supply and consumption demand of gold and soybeans; third, the price of crude oil has an impact on the demand and profit rate of biofuels; fourth, the role of gold as a hedge and safe haven asset

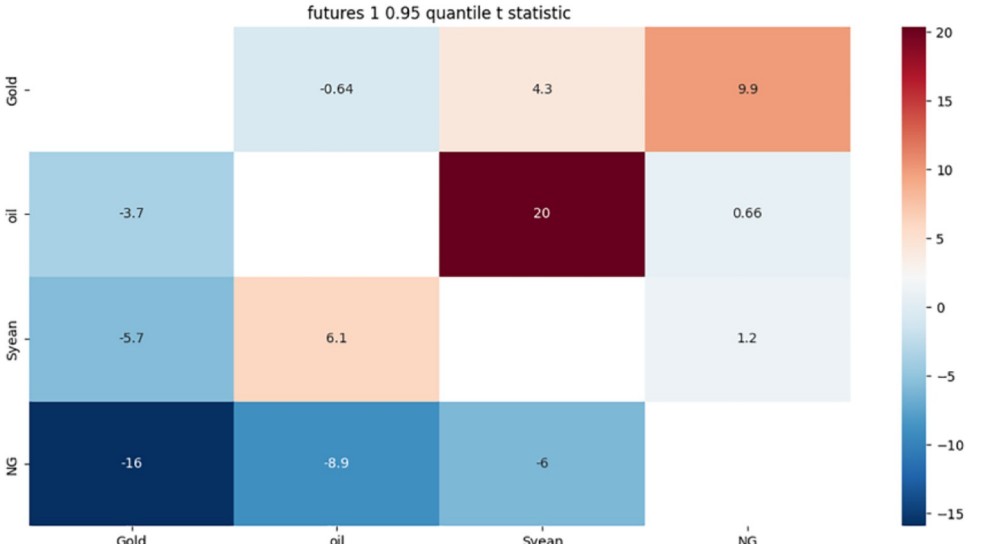

**Fig 3. 1-month approximate component 0.95 quantile risk correlation heat map.**

under inflation has an impact. These factors will lead to the same direction or reverse changes in the prices of the three, thereby increasing the risk transmission and amplification among the three.

There are two main reasons for the strong one-way risk correlation between gold and natural gas and crude oil: first, the global economic and political instability leads to the increase in demand for gold as a safe-haven asset, affecting the cost and demand of energy; second, the replacement of gold, natural gas and crude oil as international reserve assets leads to the adjustment of reserve structure. These factors make the price of gold highly sensitive to the changes of natural gas and crude oil prices, while the price of natural gas and crude oil may not have obvious feedback to the changes of gold prices.

Both crude oil and soybean have a strong one-way risk correlation to natural gas. The reasons for this phenomenon may be related to the following three aspects: First, as substitutes for natural gas, crude oil and soybean affect the demand for natural gas; Second, as important commodities in international trade, crude oil and soybean affect the exchange rate of US dollar and the international market price of natural gas; Third, as important raw materials in energy and agriculture, crude oil and soybeans have affected the production cost and demand for natural gas in these fields. These factors lead to the emergence of a one-way risk correlation between crude oil and soybean and natural gas.

Natural gas has a strong one-way risk correlation with gold. The reasons for this phenomenon may be related to the following three aspects: First, rising natural gas prices will lead to rising inflation expectations, which will push up the price of gold as an inflation hedge asset; Second, both the price of natural gas and gold are affected by the exchange rate of the US dollar. When the US dollar depreciates, the price of these two commodities will rise, and when the US dollar appreciates, the price of these two commodities will fall; Third, both natural gas and gold are one of the safe-haven choices for investors, and when there is a global economic or political crisis, the prices of these two commodities may rise in tandem. These factors lead to the occurrence of the risk correlation between natural gas and gold.

**4.4.2 Risk correlation identification of futures market with 1-month term based on detail component.** From Fig 4 we can see:

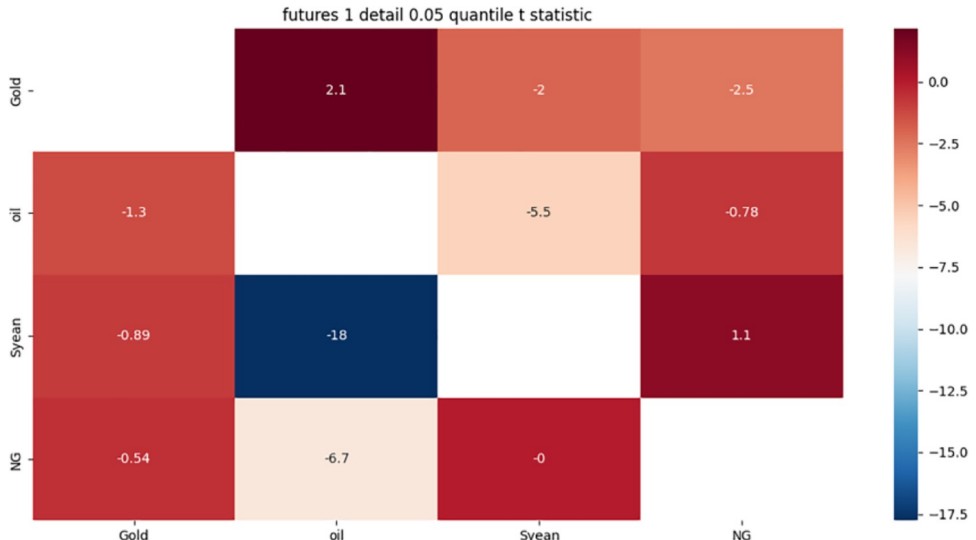

**Fig 4. One-month detail component 0.05 quantile risk correlation heat map.**

There is a strong two-way risk correlation between crude oil and soybeans. The reasons for this phenomenon may be related to the following three aspects: First, as important commodities in international trade, the prices of crude oil and soybeans are affected by the exchange rate of the US dollar. When the US dollar depreciates, the prices of these two commodities will rise, and when the US dollar appreciates, the prices of these two commodities will fall; Second, crude oil is important energy and raw material for soybean production, transportation, and processing, and its price changes will affect the cost and profit of soybeans when the price of crude oil rises, the market price of soybeans will also rise when the price of crude oil falls, the market price of soybeans will also fall; Third, soybean is one of the main raw materials of biodiesel, its price changes will affect the supply and demand of biodiesel and competitiveness when the price of soybeans rises, the demand for biodiesel will decrease, when the price of soybeans falls, the demand for biodiesel will increase. These factors lead to the emergence of a bidirectional risk correlation between crude oil and soybean.

Crude oil has a strong unidirectional risk correlation with natural gas. The reasons for this phenomenon may be related to the following three aspects: First, the impact of global economy and politics, when the global economy slows down or there is a crisis, the energy demand will decrease, leading to a decline in the price of crude oil and natural gas, when the global economy recovers or there is tension, the energy demand will increase, pushing up the price of crude oil, but the price of natural gas may not fully follow, because the supply of natural gas is relatively sufficient. And is affected by regional markets and seasonal factors; Second, the impact of chemical raw materials, when the price of crude oil falls, it will stimulate the demand for chemical products, thereby increasing the demand and price of chemical products using natural gas as raw materials, when the price of crude oil rises, it will inhibit the demand for chemical products, thereby reducing the demand and price of chemical products using natural gas as raw materials; The third is the impact of the power industry, when the price of crude oil falls, it will increase the opening probability of the oil generator set, thereby reducing the opening probability and demand of the natural gas generator set, when the price of crude oil rises, it will reduce the opening probability of the oil generator set, thereby increasing the opening probability and demand of the natural gas generator set. These factors lead to the emergence of one-way risk correlation between crude oil and natural gas.

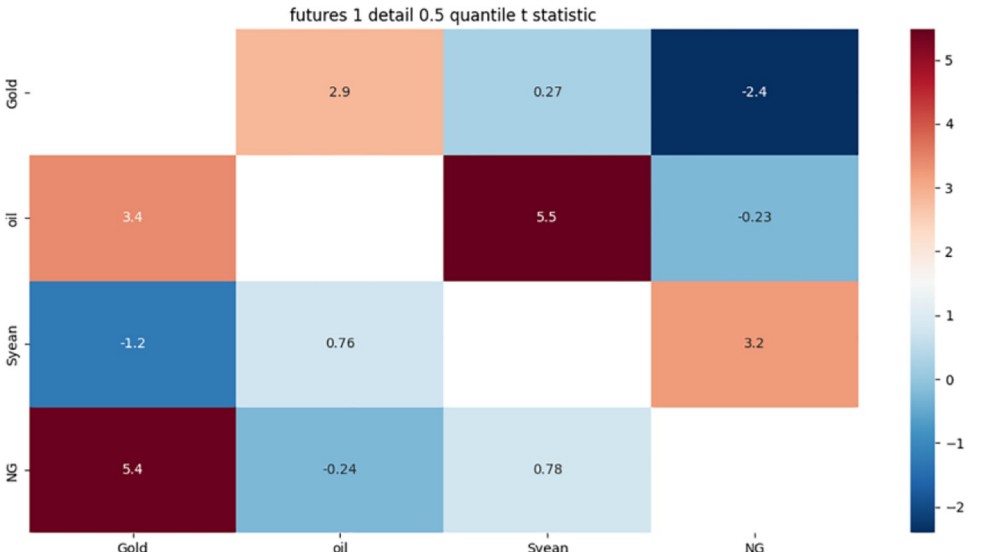

**Fig 5. 1-month detail 0.5 quantile risk correlation heat map.**

Crude oil, soybeans, and natural gas have a strong one-way risk correlation to gold, and the reasons for this phenomenon may be related to the following two aspects: One is the impact of the exchange rate of the US dollar. When the US dollar depreciates, it will increase the purchasing power of non-US consumers for crude oil, soybeans and natural gas, thus pushing up the prices of these three commodities, while reducing the negative correlation between gold and the US dollar. When the US dollar appreciates, it will reduce the purchasing power of non-US consumers for crude oil, soybeans and natural gas, thus suppressing the prices of these three commodities. At the same time, increasing the positive correlation of gold with the dollar; The second is the impact of biodiesel, when the price of crude oil, soybeans or natural gas falls, it will improve the competitiveness of biodiesel and petroleum diesel, thus stimulating the demand for biodiesel, while reducing the price of gold as an energy commodity, when the price of crude oil, soybeans or natural gas rises, it will reduce the competitiveness of biodiesel and petroleum diesel, thus inhibiting the demand for biodiesel. At the same time, increase the price of gold as an energy commodity. These factors have led to a one-way risk correlation between crude oil, soybeans and natural gas for gold.

Fig 5 shows:

There is a strong bidirectional risk correlation between gold, crude oil and natural gas, and the reason for this phenomenon may be affected by biodiesel, when the price of gold, crude oil or natural gas falls, it will improve the competitiveness of biodiesel and petroleum diesel, thus stimulating the demand for biodiesel, while reducing the price of gold as an energy commodity, when the price of gold, crude oil or natural gas rises, it will increase the demand for biodiesel. Will reduce the competitiveness of biodiesel and petroleum diesel, thereby inhibiting the demand for biodiesel, while increasing the price of gold as an energy commodity. These factors have led to a two-way risk correlation between gold, crude oil and natural gas.

There is a strong one-way risk correlation between soybeans and crude oil. The reasons for this phenomenon may be related to the following two aspects: First, when the price of soybeans falls, it will improve the competitiveness of biodiesel and petroleum diesel, thus stimulating the demand for biodiesel, and then pushing up the demand and price of crude oil; when the price of soybeans rises, it will reduce the competitiveness of biodiesel and petroleum diesel.

Thus inhibiting the demand for biodiesel, and then lowering the demand and price of crude oil; Second, the impact of the US dollar exchange rate: when the US dollar depreciates, it will increase the purchasing power of non-US consumers for soybeans, thus pushing up the price of soybeans; at the same time, it will reduce the negative correlation between gold and the US dollar, thus reducing the demand for gold as a hedge, and then increasing the demand for other risky assets such as crude oil; when the US dollar rises, it will reduce the purchasing power of non-US consumers for soybeans. Thus inhibiting the price of soybeans, while increasing the positive correlation between gold and the US dollar, thereby increasing the demand for gold as a hedge, and thereby reducing the demand for other risky assets such as crude oil. These factors have led to a one-way risk correlation between soybeans and crude oil.

There is a strong unidirectional risk correlation between natural gas and soybean. The reasons for this phenomenon may be related to the following two aspects: When the US dollar depreciates, it will increase the purchasing power of non-US consumers for natural gas and soybeans, thus pushing up the prices of these two commodities; at the same time, it will reduce the negative correlation between gold and the US dollar, thus reducing the demand for gold as a hedge, and then increasing the demand for other risky assets such as crude oil; when the US dollar rises, it will reduce the purchasing power of non-US consumers for natural gas and soybeans. Thus suppressing the price of these two commodities, while increasing the positive correlation between gold and the US dollar, thereby increasing the safe-haven demand for gold, and thereby reducing the demand for other risky assets such as crude oil; Second, when the price of natural gas falls, it will improve the competitiveness of biodiesel and petroleum diesel, thus stimulating the demand for biodiesel, and then pushing up the demand and price of soybeans, when the price of natural gas rises, it will reduce the competitiveness of biodiesel and petroleum diesel, thereby suppressing the demand for biodiesel, and then depressing the demand and price of soybeans. These factors have led to the emergence of a one-way risk association between natural gas and soybeans.

Fig 6 shows:

There is a strong bidirectional risk correlation between gold and natural gas. The reason for this phenomenon may be that when the price of gold or natural gas falls, it will improve the

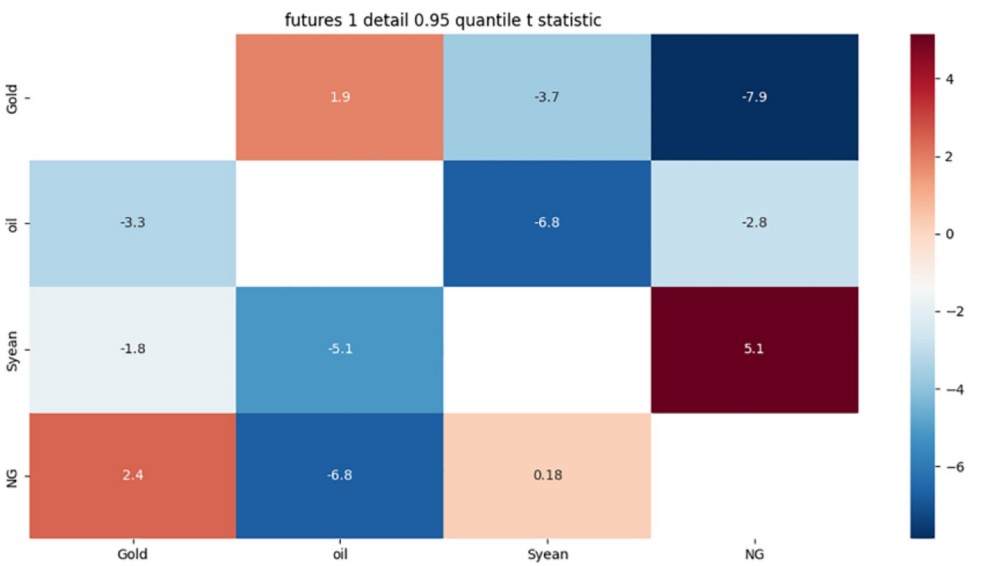

**Fig 6. 1-month detail component 0.95 quantile risk correlation heat map.**

competitiveness of biodiesel and petroleum diesel, thus stimulating the demand for biodiesel, while pushing up the demand and price of gold or natural gas. When the price of gold or natural gas rises, it will reduce the competitiveness of biodiesel and petroleum diesel. Thus suppressing the demand for biodiesel while depressing the demand and price of gold or natural gas. These factors have led to a two-way risk correlation between gold and natural gas.

There is a strong one-way risk correlation between gold and crude oil, and the reasons for this phenomenon may be related to the following two aspects: The first is the impact of inflation. When the price of crude oil rises, it will lead to the increase of production costs and the rise of consumer prices, which will lead to the rise of inflation expectations. Inflation will lead to the decline of currency purchasing power and the reduction of interest rates, which will stimulate investors to seek hedging means and increase the demand for gold. The second is the impact of geopolitical risk, when the global economic or political crisis or tension, energy or hedge demand will increase, thus pushing up the price of gold and crude oil. In particular, conflicts or unrest in the Middle East will affect the supply and stability of oil, which will lead to fluctuations in crude oil prices. Under such circumstances, investors would seek safe assets to hedge against risk, which in turn increases the demand for gold. These factors have led to a one-way risk correlation between gold and crude oil.

There is a strong bidirectional risk correlation between crude oil and soybean natural gas. The reason for this phenomenon may be that when the price of crude oil or soybean natural gas falls, it will improve the competitiveness of biodiesel and petroleum diesel, thus stimulating the demand for biodiesel and pushing up the demand and price of crude oil or soybean natural gas. When the price of crude oil or soybean natural gas rises, the demand and price of crude oil or soybean natural gas will increase. Will reduce the competitiveness of biodiesel and petroleum diesel, thereby inhibiting the demand for biodiesel, while depressing the demand and price of crude oil or soybean natural gas. These factors lead to a two-way risk correlation between crude oil and soybean natural gas.

There is a strong one-way risk correlation between soybean and gold, because both of them are international commodities affected by multiple factors such as changes in the global market environment, substitutability and complementarity, as well as market mechanism and behavior factors. Under extreme risk conditions, the price fluctuation of soybean will have a significant impact on the price of gold, while the reverse is not the case.

There is a strong one-way risk correlation between natural gas and soybeans. Due to the influence of multiple factors such as changes in global market environment, substitution and complementarity, as well as market mechanism and behavior factors, natural gas and soybeans, as international energy and agricultural products, show a strong one-way risk correlation under 0.95 quantile during the end of January 2009 to the end of March 2023. This means that in the case of extreme risk, the price fluctuation of natural gas will significantly affect the price of soybeans, but the price fluctuation of soybeans will not significantly affect the price of natural gas.

## 4.5 Identification of risk associations in the futures market with a six-month term

### 4.5.1 Risk association identification of the futures market based on the approximate component of the 6-month term. Fig 7 shows:

The strong bidirectional risk correlation between gold and natural gas may be due to the fact that first, both are priced in US dollars, and second, gold prices tend to rise when inflation increases or exchange rate fluctuations occur, and vice versa. As a kind of clean energy, the price of natural gas is related to the price of oil, environmental protection policies,

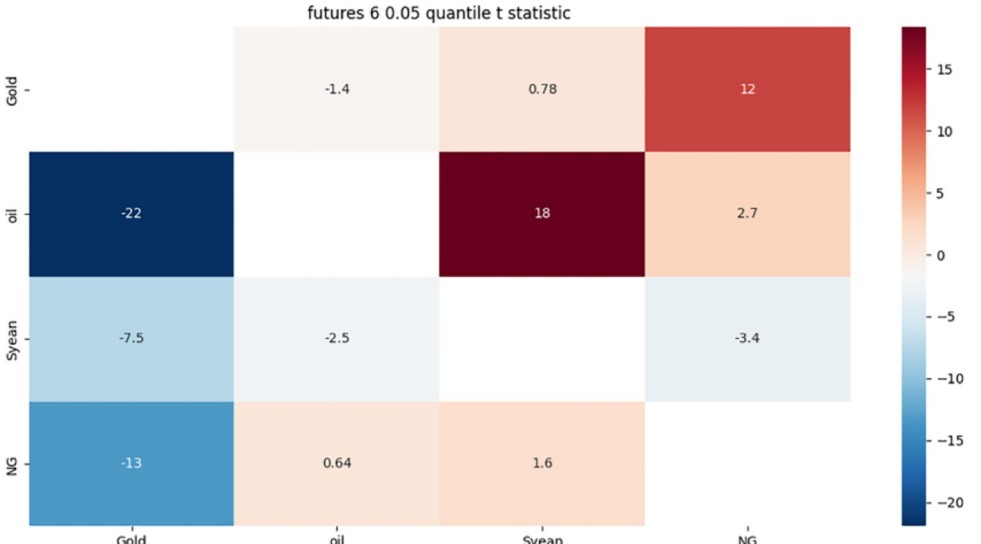

**Fig 7. Risk association heat map of approximate component 0.05 quantile at 6 months.**

technological progress and other factors. When oil prices rise or environmental protection requirements increase, the price of natural gas will also rise, and vice versa.

Both gold and crude soybeans have a strong one-way risk correlation, mainly due to their close correlation to the dollar exchange rate.

The strong bidirectional risk association between crude oil and soybeans can be caused by a variety of factors, including their correlation in energy markets, their exposure to international market and political factors, and their alternative or complementary relationship with other commodities. These factors can lead to a pattern of positive or negative correlation between the price fluctuations of the two, resulting in a strong bidirectional risk association.

Natural gas has a strong one-way risk association with crude oil soybeans, which can be caused by a variety of factors, including the impact of both on climate change, international trade policies and global economic growth, as well as the substitution or complementarity of both with other commodities. These factors will lead to a positive or negative correlation between the price fluctuations of the two commodities, thus forming a strong one-way risk association.

It can be seen from Fig 8:

There is a significant two-way risk association between gold and soybeans, which can be caused by a variety of factors, including the nature of both as safe haven assets, the impact of both on the exchange rate of the US dollar, and the existence of a substitute or complementary relationship between both and other commodities. These factors can lead to a pattern of positive or negative correlation between the price fluctuations of the two commodities, thus forming a significant bidirectional risk association.

Soybeans have a strong one-way risk association with crude oil, which can result from a variety of factors, including their exposure to global market and political events, their correlation with biofuels, and their correlation with the U.S. dollar exchange rate. These factors can lead to a pattern of positive or negative correlation between the price movements of the two, resulting in a strong one-way risk association.

Natural gas has a strong one-way risk association with both gold and crude oil, which can result from a variety of factors, including their relationship to global economic growth, the

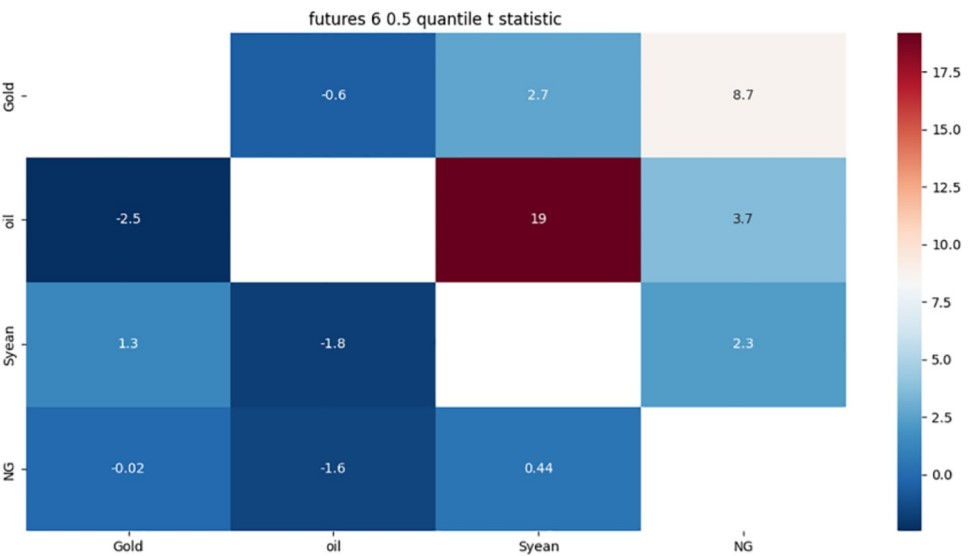

**Fig 8. 6-month approximate component 0.5 quantile risk correlation heat map.**

exchange rate of the U.S. dollar, and other energy commodities. These factors can lead to a pattern of positive or negative correlation between the price fluctuations of natural gas and crude oil, thus forming a strong one-way risk association.

From Fig 9:

There is a strong two-way risk association between gold and natural gas, which can be caused by a number of factors, including the relationship between the two and international trade policies, global economic growth, and the exchange rate of the US dollar. These factors can lead to a pattern of positive or negative correlation between the price fluctuations of the two, thus forming a strong bidirectional risk association.

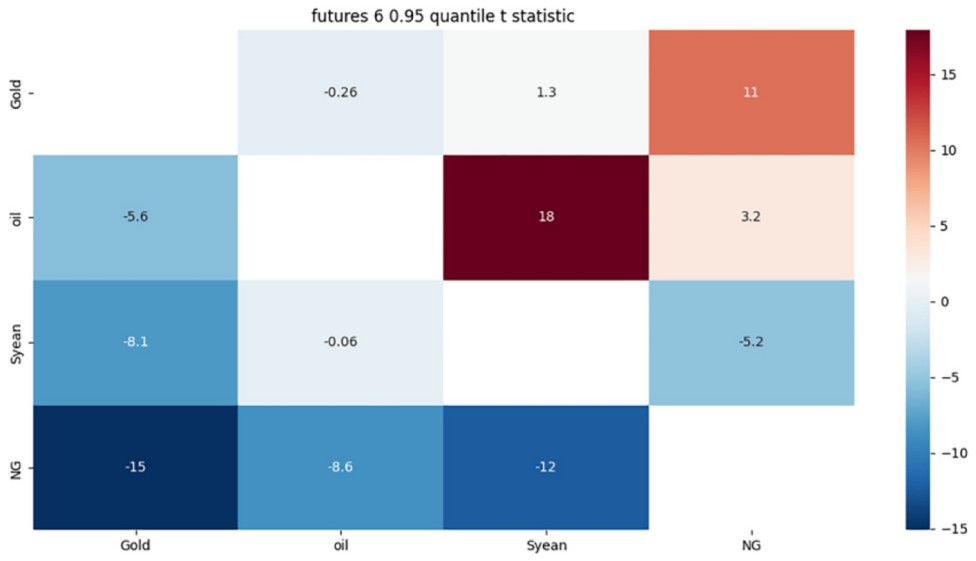

**Fig 9. 6-month approximate component 0.95 quantile risk correlation heat map.**

There is a strong one-way risk correlation between gold and crude oil soybeans, possibly due to their relationship with global supply and demand, as well as their substitution or complementarity with other commodities. These factors will lead to a positive or negative correlation between the price fluctuations of the two commodities, thus forming a strong one-way risk association.

The strong two-way risk association between crude oil and natural gas can be caused by a variety of factors, including the relationship between the two with global supply and demand imbalances, international political and geopolitical risks, and a strong US dollar exchange rate. These factors will lead to a pattern of positive or negative correlation between the price fluctuations of crude oil and natural gas, thus forming a strong bidirectional risk correlation.

The strong bidirectional risk association between soybeans and natural gas can result from a variety of factors, including their relationship to biofuel production and consumption, global food security and the exchange rate of the US dollar. These factors can lead to a pattern of positive or negative correlation in price fluctuations between the two, resulting in a strong bidirectional risk association.

Soybeans have a strong one-way risk association with crude oil, which can be caused by a variety of factors, including their relationship to biodiesel production and consumption, global food security and the US dollar exchange rate. These factors can lead to a pattern of positive or negative correlation between the price fluctuations of the two, resulting in a strong one-way risk association.

**4.5.2 Risk association identification in the futures market based on the 6-month term of detail components.** From Fig 10:

The strong bidirectional risk correlation between crude oil and soybean is mainly due to the fact that crude oil and soybean, as international commodities, are not only affected by themselves, but also by the international financial market.

Gold has a strong one-way risk correlation with crude oil, mainly due to the rise in crude oil prices, leading to the intensification of related inflation, the increase in economic uncertainty, leading to violent investor sentiment fluctuations.

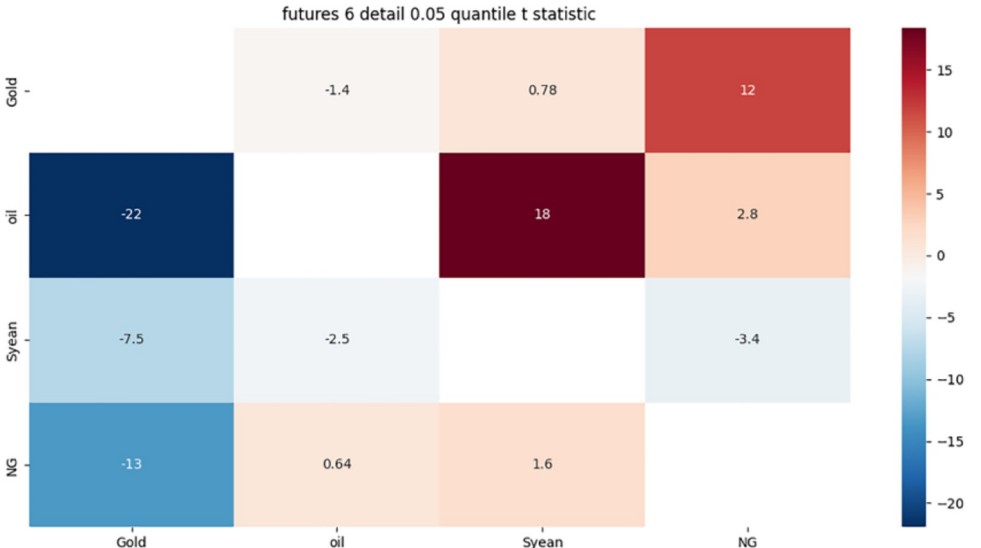

**Fig 10. 6-month detail component 0.05 quantile risk correlation heat map.**

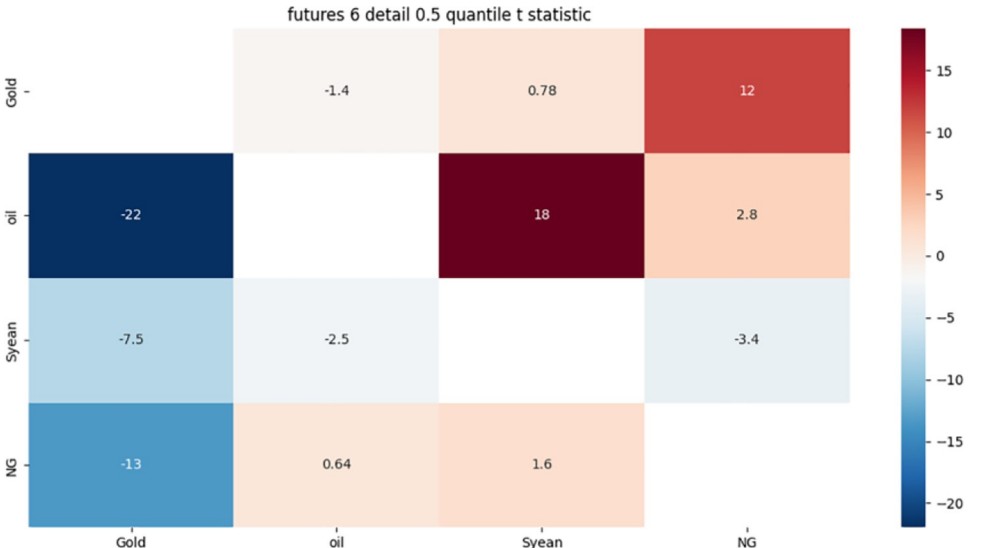

**Fig 11. 6-month detail component 0.5 quantile risk correlation heat map.**

Crude oil has a strong one-way risk association with natural gas, mainly due to the exchange rate fluctuations of the relevant crude oil supply and demand countries and the seasonality, regionalism and substitution of natural gas.

Soybean has a strong one-way risk association with gold, which may affect the price trend of gold due to consumers' expectations of the market.

Natural gas has a strong one-way risk correlation to gold, which is influenced by international geopolitics. International geopolitical events tend to trigger uncertainty and risk aversion in the market.

From Fig 11:

Gold has a strong one-way risk association with crude oil, possibly because gold is a safe haven asset, while crude oil is affected by weather, policy, as well as supply and demand.

The strong bidirectional risk correlation between crude oil and soybeans may be due to the fact that crude oil and soybeans, as important international general-purpose raw materials, are affected by cost, substitution and supply and demand.

There is a strong one-way risk correlation between crude oil and gold, which may be due to the one-way risk correlation caused by the investment characteristics of crude oil and gold.

The strong risk association between natural gas and gold may be due to the fact that both are dollar-denominated and show a strong correlation with the US dollar exchange rate.

Fig 12 shows:

There is a strong two-way risk correlation between gold and soybeans, which can be affected by slowing global economic conditions, inflationary pressures, monetary policy, and international events.

There is a strong one-way risk correlation between crude oil and gold, which may be due to the market's lower expectations of economic growth, rising inflation, lower interest rates and other factors that lead to gold as a safe haven asset, and at the same time lead to the rise in crude oil prices.

Soybean has a strong one-way risk association with crude oil, which may be due to some common factors between soybean and crude oil, such as global economic growth, political

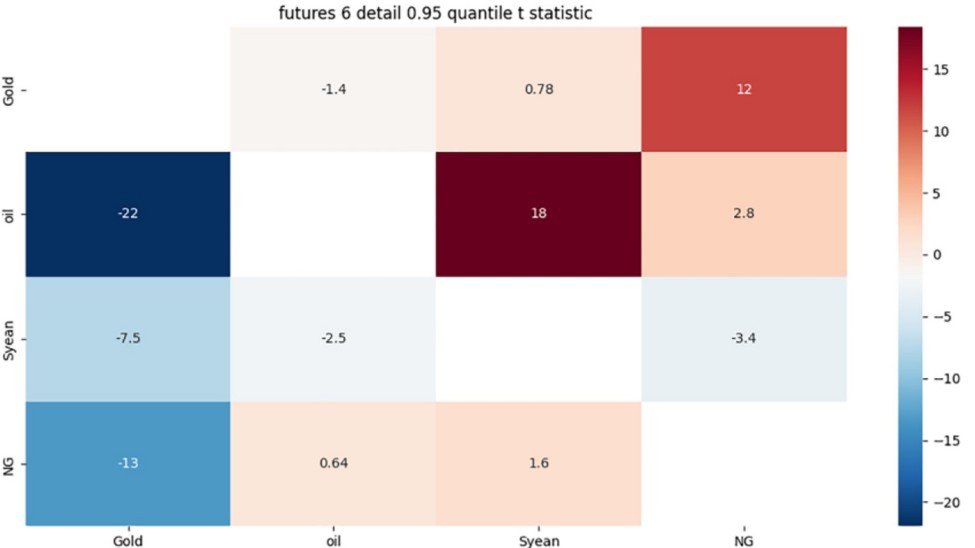

**Fig 12. 6-month detail component 0.95 quantile risk correlation heat map.**

stability, weather and other factors. These factors may affect the supply and demand relationship between soybean and crude oil, resulting in a strong one-way risk association between them.

The reason for the strong one-way risk correlation between natural gas and gold is that both natural gas and gold are important commodities whose prices are affected by a variety of factors, such as global economic conditions, political stability, market supply and demand, and so on. In addition, the correlation between natural gas and gold can also be affected by other factors, such as inflation rates, exchange rate changes, and so on. In summary, the one-way risk correlation between natural gas and gold is caused by a combination of factors.

## 5 Robustness test

In the robustness test, this paper adopts the method of combining substitution variables and rolling window estimation to conduct robustness test. The specific results are shown in Tables 7–18, and it can be found whether there is a risk correlation among gold, crude oil, soybean and natural gas in 1-month futures contracts and 6-month futures contracts under different quantiles, and how large the risk correlation is.

**Table 7. Rolling window estimation of 0.05 quartile approximate component of Granger causal t statistic value matrix for 1 month futures contracts of gold, crude oil, soybean and natural gas.**

| Variables | Gold | oil | Sean | NG |
|---|---|---|---|---|
| Gold | - | 0.47 | 2.25 | 12.88 |
| oil | 22.25 | - | 17.42 | 7.36 |
| Sean | 2.00 | 6.87 | - | 0.21 |
| NG | 6.40 | 2.05 | 0.019 | - |

Note: Gold stands for gold futures price, oil stands for Brent crude futures price, Syean stands for soybean futures price, NG stands for natural gas futures price, the same below.

**Table 8. Rolling window estimate of Granger causal t statistic value matrix for 0.5 quartile approximate component of 1-month futures contracts for gold, crude oil, soybean and natural gas.**

| Variables | Gold | oil | Sean | NG |
|---|---|---|---|---|
| Gold | - | 0.26 | 3.52 | 10.55 |
| oil | 6.06 | - | 20.05 | 3.18 |
| Sean | 2.24 | 6.19 | - | 0.44 |
| NG | 0.76 | 1.93 | 0.53 | - |

Note: Gold stands for gold futures price, oil stands for Brent crude futures price, Syean stands for soybean futures price, NG stands for natural gas futures price, the same below.

The correctness of the conclusions of one-month term, 6-month term approximate component and detail component under 0.05 quantile, 0.5 quantile and 0.95 quantile analysis is verified from three conclusions.

# 6 Conclusions and policy recommendations

## 6.1 Comparison with literature findings

In order to better illustrate the contribution of this paper, five literatures with the highest similarity to the research content of this paper are selected for comparison, which are Liu et al. (2016), Wang et al. (2017), Ma Yanran et al. (2020), Liu Yinglin et al. (2019) and Zhang Dayong et al. (2018). The research methods, research objects, research areas, and research results are discussed as follows:

From the perspective of research methods: the five literatures all use different methods and models to analyze the data, such as MGARCH model, DFA method, CVaR model, RSI method, MRCSB method, DCC model, ROM method and wavelet transform quantile Granger

**Table 9. Rolling window estimate of Granger causal t statistic value matrix of 0.95 fractional approximate component for 1-month futures contracts of gold, crude oil, soybean and natural gas.**

| variable | Gold | oil | Sean | NG |
|---|---|---|---|---|
| Gold | - | 1.05 | 3.91 | 8.76 |
| oil | 3.80 | - | 20.37 | 0.66 |
| Sean | 5.34 | 6.14 | - | 1.19 |
| NG | 15.79 | 8.87 | 5.96 | - |

Note: Gold stands for gold futures price, oil stands for Brent crude futures price, Syean stands for soybean futures price, NG stands for natural gas futures price, the same below.

**Table 10. The approximate component of 0.05 quantile for 6-month futures contracts of gold, crude oil, soybean and natural gas Granger causal t statistic value matrix rolling window estimation.**

| Variables | Gold | oil | Sean | NG |
|---|---|---|---|---|
| Gold | - | 1.335 | 1.605 | 10.86 |
| oil | 20.83 | - | 18.37 | 2.80 |
| Sean | 7.46 | 2.49 | - | 3.35 |
| NG | 12.00 | 0.64 | 1.64 | - |

Note: Gold stands for gold futures price, oil stands for Brent crude futures price, Syean stands for soybean futures price, NG stands for natural gas futures price, the same below.

 

**Table 11. Rolling window estimate of Granger causal t statistic value matrix for 0.5 quartile approximate component of 6-month futures contracts for gold, crude oil, soybean and natural gas.**

| Variables | Gold | oil | Sean | NG |
|---|---|---|---|---|
| Gold | - | 0.43 | 2.99 | 8.52 |
| oil | 2.45 | - | 19.19 | 3.71 |
| Sean | 1.36 | 1.84 | - | 2.33 |
| NG | 0.02 | 1.56 | 0.45 | - |

Note: Gold stands for gold futures price, oil stands for Brent crude futures price, Syean stands for soybean futures price, NG stands for natural gas futures price, the same below.

causality test. Each of these methods and models has its advantages and disadvantages. For example, the MGARCH model can consider the volatility spillover and contagion effect between different varieties, but cannot distinguish the impact of positive and negative shocks. The DFA method can decompose the structural components of China's futures market, but can not capture the nonlinear relationship; CVaR model can measure the extreme risk level, but can not reflect the direction of risk spillover. RSI method can measure the intensity of risk spillover, but can not reflect the persistence of risk spillover. The MRCSB method can identify the time and degree of the impact of international crude oil price fluctuations on China's commodity futures but can not consider the multiple correlation structure; the DCC model can capture the dynamic conditional correlation but can not distinguish the upstream and downstream risk spillovers; ROM method can measure the dynamic risk spillover effect, but can not reflect the asymmetry of risk spillover. The wavelet transform quantile Granger causality test adopted in this paper can identify different frequency bands and different degrees of risk correlation.

**Table 12. Rolling window estimate of Granger causality T-statistic value matrix for 6-month futures contracts of gold, crude oil, soybean and natural gas with 0.95 quantile approximate component.**

| Variables | Gold | oil | Sean | NG |
|---|---|---|---|---|
| Gold | - | 0.29 | 1.22 | 9.84 |
| oil | 5.86 | - | 17.92 | 3.18 |
| Sean | 7.90 | 0.069 | - | 5.22 |
| NG | 14.39 | 8.63 | 12.35 | - |

Note: Gold stands for gold futures price, oil stands for Brent crude futures price, Syean stands for soybean futures price, NG stands for natural gas futures price, the same below.

**Table 13. 0.05 quartile detail components of gold, crude oil, soybean and natural gas 1-month futures contracts Granger causal t statistic value matrix rolling window estimation.**

| Variables | Gold | oil | Sean | NG |
|---|---|---|---|---|
| Gold | - | 0.57 | 2.49 | 2.00 |
| oil | 1.35 | - | 5.56 | 0.78 |
| Sean | 0.95 | 17.76 | - | 1.10 |
| NG | 0.54 | 6.66 | 0.002 | - |

Note: Gold stands for gold futures price, oil stands for Brent crude futures price, Syean stands for soybean futures price, NG stands for natural gas futures price, the same below.

**Table 14. 0.5 quantile detail components of 1-month futures contracts for gold, crude oil, soybean and natural gas Granger causal t statistic value matrix rolling window estimation.**

| Variables | Gold | oil | Sean | NG |
|---|---|---|---|---|
| Gold | - | 2.99 | 0.24 | 2.28 |
| oil | 3.47 | - | 5.49 | 0.24 |
| Sean | 1.27 | 0.77 | - | 3.20 |
| NG | 5.30 | 0.25 | 0.79 | - |

Note: Gold stands for gold futures price, oil stands for Brent crude futures price, Syean stands for soybean futures price, NG stands for natural gas futures price, the same below.

From the perspective of research objects, the five papers all involve different varieties of China's futures market, such as crude oil, rubber, copper, aluminum, corn and soybean, but their concerns are different. Liu et al. (2016) and Wang et al. (2017) both analyzed the volatility structure and dynamic characteristics of China's futures market as a whole. While Ma Yanran et al. (2020), Liu Yinglin et al. (2019) and Zhang Dayong et al. (2018) studied the risk spillover effects between domestic crude oil futures and other financial assets, international crude oil prices and Chinese commodity futures, and Chinese crude oil futures and other financial assets respectively. This paper is based on wavelet transform—quantile Granger causality test to identify the risk correlation among precious metal futures market, crude oil futures market, grain futures market and common energy futures market.

From the perspective of the research area: Liu et al. (2016), Wang et al. (2017), Ma Yanran et al. (2020), Liu Yinglin et al. (2019), Zhang Dayong et al. (2018) all studied the risk correlation among China futures variety markets. This paper studies the risk correlation identification of the precious metal futures market, the crude oil futures market, the grain futures market

**Table 15. 0.95 quantile detail components of 1-month futures contracts for gold, crude oil, soybeans and natural gas Granger causal t statistic value matrix rolling window estimate.**

| Variables | Gold | oil | Sean | NG |
|---|---|---|---|---|
| Gold | - | 2.29 | 2.72 | 7.88 |
| oil | 2.95 | - | 6.75 | 2.84 |
| Sean | 0.93 | 5.13 | - | 5.15 |
| NG | 2.57 | 6.76 | 0.18 | - |

Note: Gold stands for gold futures price, oil stands for Brent crude futures price, Syean stands for soybean futures price, NG stands for natural gas futures price, the same below.

**Table 16. 0.05 quantile detail components of 6-month futures contracts for gold, crude oil, soybean and natural gas Granger causal t statistic value matrix rolling window estimation.**

| Variables | Gold | oil | Sean | NG |
|---|---|---|---|---|
| Gold | - | 0.44 | 2.16 | 1.01 |
| oil | 3.68 | - | 4.98 | 1.45 |
| Sean | 1.42 | 3.60 | - | 1.80 |
| NG | 1.31 | 9.38 | 0.84 | - |

Note: Gold stands for gold futures price, oil stands for Brent crude futures price, Syean stands for soybean futures price, NG stands for natural gas futures price, the same below.

Table 17. **Estimates of 0.5 quantile detail components of 6-month futures contracts for gold, crude oil, soybean and natural gas by Granger causal t statistic matrix rolling window.**

| Variables | Gold | oil | Sean | NG |
|---|---|---|---|---|
| Gold | - | 3.63 | 1.23 | 3.42 |
| oil | 3.73 | - | 13.09 | 0.07 |
| Sean | 0.01 | 3.83 | - | 1.70 |
| NG | 0.10 | 2.20 | 0.31 | - |

Note: Gold stands for gold futures price, oil stands for Brent crude futures price, Syean stands for soybean futures price, NG stands for natural gas futures price, the same below.

and the common energy futures market of the United States with 1 month and 6 month contracts

From the research results, the five papers have obtained some valuable results, such as the Chinese futures market has significant volatility accumulation, leverage effect and persistence phenomenon; There is a common potential factor and a specific factor for each variety in China's futures market; There is a bi-directional extreme risk spillover effect between domestic crude oil futures and other financial assets, and it is affected by time period and market sentiment; The fluctuation of international crude oil prices has a significant positive impact on China's commodity futures, and there are differences in different varieties; There is a bidirectional dynamic risk spillover effect between China's crude oil futures and other financial assets, and it changes under different market sentiment; This paper draws the conclusion that there are risk correlations in different degrees and directions among precious metal futures market, crude oil futures market, grain futures market and common energy futures market, and they are stronger at high frequency band and high number.

## 6.2 Conclusion

From the above analysis, it can be concluded that:

First, there is a strong bidirectional risk correlation between gold and natural gas regardless of 1-month contract or 6-month contract.

Second, in the 1-month futures contract, there is an indirect bidirectional risk correlation between gold and crude oil, while in the 6-month futures, gold has a strong unidirectional risk correlation with crude oil, that is, the change of gold price will affect the change of crude oil price.

Third, in the 1-month futures contract, there is a strong direct bidirectional risk correlation between crude oil and soybeans and natural gas, while in the 6-month futures, there is a strong

Table 18. **0.95 quantile detail components of 6-month futures contracts for gold, crude oil, soybeans and natural gas Granger causal t statistic value matrix rolling window estimates.**

| Variables | Gold | oil | Sean | NG |
|---|---|---|---|---|
| Gold | - | 6.21 | 3.39 | 9.48 |
| oil | 2.61 | - | 3.43 | 0.76 |
| Sean | 0.40 | 0.47 | - | 1.36 |
| NG | 1.64 | 0.82 | 1.02 | - |

Note: Gold stands for gold futures price, oil stands for Brent crude futures price, Syean stands for soybean futures price, NG stands for natural gas futures price, the same below.

bidirectional risk correlation between crude oil and soybeans, but natural gas has a strong unidirectional risk correlation between crude oil and soybeans.

Fourth, there is a bi-directional risk association between gold and soybeans in both 1-month and 6-month futures contracts, but it is weaker at the 0.05 quantile and stronger at the 0.5 and 0.95 quantiles. This shows that the correlation between gold and soybean price movements increases with the degree of price volatility.

Fifth, at 0.95 quantile, there is a strong bidirectional or unidirectional risk association between all varieties.

Sixth, in the two futures varieties of 1-month contract and 6-month contract, there is a strong bidirectional risk correlation between crude oil and soybeans, that is, the price changes of crude oil and soybeans will affect each other.

Seventh, in the one-month contract, there is a strong one-way risk correlation between crude oil and natural gas, that is, the price change of crude oil will affect the price change of natural gas, but the reverse is not true.

Eighth, in the six-month contract, gold has a strong one-way risk correlation with crude oil, that is, the price change of gold will affect the price change of crude oil, but the vice versa is not true.

Ninth, in the 1-month contract and 6-month contract, soybean has a strong one-way risk correlation with gold, that is, the price change of soybean will affect the price change of gold, but the vice versa is not true.

Tenth, at 0.5 quantile and 0.95 quantile, there is a strong bidirectional risk correlation between gold and natural gas, that is, the price changes of gold and natural gas will affect each other.

Eleventh, at 0.95 quantile, there is a strong bidirectional risk correlation between crude oil, soybeans and natural gas, that is, the price changes of crude oil, soybeans and natural gas will affect each other.

As can be seen from the above conclusions, the above conclusions indicate that this paper uses the wavelet transform-quantile Granger causality test to identify and measure the risk correlation of four major varieties (gold, crude oil, soybean, and natural gas) in China's futures market and reveals the differences and changes in different frequency bands and quantile levels. It verifies the research purpose proposed in the introduction of this paper and answers the questions about the risk correlation among different futures varieties and the change with different frequency bands and quantiles.

This study finds the above points, which are similar to the previous ones, such as the accumulation of volatility, leverage effect and persistence phenomenon in China's futures market; There is a common potential factor and a unique factor of each variety in China's futures market; There is a bi-directional extreme risk spillover effect between domestic crude oil futures and other financial assets, and it is affected by time period and market sentiment; The fluctuation of international crude oil prices has a significant positive impact on China's commodity futures, and there are differences in different varieties; There is a bidirectional dynamic risk spillover effect between China's crude oil futures and other financial assets, and the studies on the change under different market sentiment are different.

This paper only constructs wavelet transform and quantile Granger causality test to identify the risk correlation among precious metal futures market, crude oil futures market, grain futures market and common energy futures market, providing a new perspective and method for the risk spillover effect of futures market, revealing the risk correlation of different varieties, different frequency bands, different degrees and different directions. However, this paper also has some limitations, such as the applicability of wavelet transform-quantile Granger causality test method. Future studies can further expand the sample range, compare the effects of

different methods, explore the risk transmission mechanism between different varieties, and introduce low-frequency macro variables with high correlation.

## 6.3 Research implications

This study provides enlightenments for relevant researchers in the following aspects:

First, different varieties, different frequency bands, different degrees and different directions of risk correlation generally exist in the futures market, which has important guiding significance for futures investors and regulators. Other researchers further explore the formation mechanism, influencing factors and dynamic changes of these risk associations, so as to improve the risk identification and management ability of relevant entities in the futures market.

Second, this paper adopts the method of wavelet transform—quantile Granger causality test, which is a novel method, which can effectively capture different frequency bands and different degrees of risk correlation. Other researchers can further compare the advantages and disadvantages of this method with other methods, as well as its applicability and robustness in different data and contexts.

Third, this paper deals with only four futures, namely gold, crude oil, soybeans and natural gas. Other researchers may further expand your research to include more futures varieties, such as copper, cotton, corn, etc., to test whether the results of this study are valid in other varieties and whether there is inter-variety heterogeneity.

Fourth, only 1 month and 6 month futures contract data are used in this paper. Other researchers may further consider longer or shorter maturities, such as 3 months, 9 months, or 12 months, to test whether the results of this study are consistent across different maturities and whether there is a term structure effect.

Fifth, this paper only considers the impact of price changes on risk correlation. Other researchers can further consider other variables that may affect risk correlation, such as trading volume, open position, leverage ratio, liquidity, etc., in order to enrich the research content and conclusions on the topic of futures risk correlation.

## 6.4 Policy recommendations

This study makes the following recommendations for the above findings:

First, in order to prevent the occurrence of systemic risks, the risk supervision of futures market should be strengthened, and effective measures such as margin system, price limit system, position limit system, trading limit system, large account position reporting system, forced closing system, forced reduction system, settlement guarantee system and risk warning system should be adopted. These measures can effectively control the leverage ratio, price fluctuations, market concentration, transaction costs and other risk factors in the futures market, and maintain the stability and health of the futures market.

Second, in order to give full play to the functions of the futures market, it is necessary to use futures and derivatives instruments for risk management and hedging, improve the price stability of agricultural products and the income level of farmers, explore innovative models such as "insurance + futures", incorporate them into the policy-based insurance system, and establish a risk-sharing system between the government and the insurance and futures markets. These measures can effectively mitigate the impact of commodity price fluctuations on farmers' income and improve their ability to resist risks and their income level.

Third, in order to promote the healthy development of the futures market, it is necessary to improve the laws and regulations of the futures market, clarify the legal provisions on the qualification of the main body of futures trading, contract design, trading rules and regulatory

responsibilities, standardize the behavior and order of the futures market, and protect the legitimate rights and interests of investors. These measures can effectively regulate the participants, products, processes, and other aspects of the futures market and improve the transparency and fairness of the futures market.

Fourth, in order to improve the efficiency and competitiveness of the futures market, innovation and opening up of the futures market should be strengthened, more new varieties, new mechanisms and new businesses should be introduced in line with market demand and national strategies, and the scope and channels of participation of domestic and foreign investors should be expanded. These measures can effectively enrich the functions and services of the futures market and enhance its vitality and attractiveness.

Fifth, in order to enhance the international influence and voice of the futures market, it is necessary to strengthen international cooperation and exchanges in the futures market, participate in the formulation of rules and standards by international organizations and institutions, and promote the pricing and settlement of domestic varieties in the international market. These measures can effectively improve the status and role of China in the international commodity market.

## Supporting information

**S1 File.**
(DOCX)

## Author Contributions

**Data curation:** Zi Qian Wu.

**Formal analysis:** Zi Qian Wu.

**Methodology:** Zi Qian Wu.

**Resources:** Zi Qian Wu.

**Software:** Zi Qian Wu.

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
