## [Decision Letter · Decision Letter 0]

10 Oct 2023

PONE-D-23-30906

Risk correlation identification of futures market based on wavelet transform and quantile Granger causality test

PLOS ONE

Dear Dr. wu,

Thank you for submitting your manuscript to PLOS ONE. After careful consideration, we feel that it has merit but does not fully meet PLOS ONE’s publication criteria as it currently stands. Therefore, we invite you to submit a revised version of the manuscript that addresses the points raised during the review process.

We look forward to receiving your revised manuscript.

Kind regards,

Chinnadurai Kathiravan

Academic Editor

PLOS ONE

   "NO"

5. We note you have included a table to which you do not refer in the text of your manuscript. Please ensure that you refer to Table 5-10 in your text; if accepted, production will need this reference to link the reader to the Table.

Reviewers' comments:

Reviewer's Responses to Questions

**Comments to the Author**

1. Is the manuscript technically sound, and do the data support the conclusions?

Reviewer #1: Partly

Reviewer #2: Yes

2. Has the statistical analysis been performed appropriately and rigorously? 

Reviewer #1: Yes

Reviewer #2: Yes

3. Have the authors made all data underlying the findings in their manuscript fully available?

Reviewer #1: Yes

Reviewer #2: No

4. Is the manuscript presented in an intelligible fashion and written in standard English?

Reviewer #1: No

Reviewer #2: Yes

5. Review Comments to the Author

Reviewer #1: 1. abstract is not clear. it is very vague. try to make is more explicit.

2. problems, need of study, gaps, and objectives are not clear. The paragraphs are very poorly drafted.

3. literature are very old, new literature must be incorporated. Literature must be enrich inline with problem of study and objectives.

4. Research methodology is not clear. it must be scientific and structured.

5. discuss the key findings with literature in a separate section.

6. include implications and future scope of the study along with limitations.

7. proof editing is require to provide coherency in the entire manuscript.

Reviewer #2: 1.The author can perform Johnson Co-integration test to analyse the variables

2.Conclusion need to be elaborate

3.Implications should be more specific

4.Gramatical errors were found

5.Latest Literature review can be added

6. PLOS authors have the option to publish the peer review history of their article (what does this mean?). If published, this will include your full peer review and any attached files.

Reviewer #1: No

Reviewer #2: No

---

## [Author Response · Author response to Decision Letter 0]

13 Oct 2023

Review Comment 1: abstract is not clear. it is very vague. try to make is more explicit.

Reply to reviewer 1：Dear reviewer, thank you very much for your questions about the abstract. According to your suggestions, I modified the abstract and the part of the abstract, adding the research purpose, method, limitations of the method and future research direction, and modified the conclusion as follows: The futures market is an important part of the financial market, which has a high degree of liquidity and leverage. However, the futures market is also faced with various risk factors, such as price fluctuations, market shocks, and supply and demand changes. In order to better determine the risk correlation among specific futures markets, this paper uses the wavelet transform and quantile Granger causality test method to identify the risk correlation in four major futures markets of gold, crude oil, soybean and natural gas in the US futures market from the end of January 2009 to the end of March 2023. It provides a new perspective and method to identify the risk correlation in futures market. The results show that the price fluctuation of futures contracts with different maturities and different quantiles has a significant impact on the risk correlation. Specifically, there is the strongest two-way risk association between gold and natural gas in 1-month and 6-month futures contracts (T-statistics of -15.94 and 10.92, respectively); In the 1-month futures contract, there is also a strong bidirectional risk correlation between crude oil, soybean and natural gas (t statistics are 6.87, 17.42, -2.05, 7.35), while in the 6-month futures contract, there is a bidirectional risk correlation between crude oil and soybean (t statistics are -2.49 and 18.374). However, natural gas has unidirectional risk association with crude oil and soybean (t statistics are 2.7 and -3.35, respectively). There is a bidirectional risk correlation between gold and soybean, that is, the risk correlation between gold and soybean increases with the increase of price fluctuation. There is a one-way risk association between gold and crude oil, soybean and gold, and crude and natural gas (the t statistic is greater than the critical value of 1.96). In addition, at 0.95 quantile, there was a strong bidirectional or unidirectional risk association among all varieties. The research results of this paper have certain reference value for the supervision, investment and risk management of the futures market. This paper uses the wavelet transform and quantile Granger causality test method to identify the risk correlation of the futures market, providing a new perspective and method for the risk correlation identification of the futures market, and uses relatively new data to ensure the effectiveness of empirical analysis. However, there are some limitations in this paper, such as the applicability of wavelet transform quantile Granger causality test. Future studies can further expand the sample scope, compare the effects of different methods, and explore the risk transmission mechanism between different varieties.

I hope the modification will do its best to meet your requirements.

Review Comment 2: problems, need of study, gaps, and objectives are not clear. The paragraphs are very poorly drafted.

Reply to reviewer 1：Dear reviewer, thank you very much for your questions about research purpose, research question and research gap. According to your suggestions, I added research question in the first paragraph of introduction, research purpose in the second paragraph of introduction, research gap in the 8th paragraph of literature review, and optimized the layout of paragraphs.

I hope the modification will do its best to meet your requirements.

Review Comment 3: literature are very old, new literature must be incorporated. Literature must be enrich inline with problem of study and objectives.

Reply to reviewer 1：Dear reviewer, thank you for pointing out that my literature review lacks recent references related to my research problems. I revised the literature review according to your suggestions, mainly including: First, 11 related literatures were added, which were as follows:

Ma Yanran, Hu Min, Zhang Dayong, et al. Extreme risk spillover of domestic crude oil futures and other financial assets [J]. Research of Environmental Economics,20,5(3):115-132.

YANG Y Y, MA Y R, HU M, et al. Extreme risk spillover between chinese and global crude oil futures[J]. Finance Research Letters, 2021, 40: 101743.

Liu Yinglin, LIU Yonghui, JU Zhuo. The impact of international crude oil price fluctuation on China's commodity futures: An analysis based on multiple correlation structure breakpoints [J]. Chinese Journal of Management Science,2019,27(2):31-40.

KANG S H, MCIVER R, YOON S M. Dynamic spillover effects among crude oil, precious metal, and agricultural commodity futures markets[J]. Energy Economics, 2017, 62: 19-32.

WU B B. The dynamics of oil on China’s commodity sectors: What can we learn from a quantile perspective? [J]. Journal of Commodity Markets, 2021, 23: 100158.

MENG J, NIE H, MO B, et al. Risk spillover effects from global crude oil market to china’s commodity sectors[J]. Energy, 2020, 202: 117208.

AHMED A D, HUO R. Volatility transmissions across international oil market, commodity futures and stock markets: Empirical evidence from China[J]. Energy Economics, 2021, 93: 104741.

LI Z H, SU Y Y. Dynamic spillovers between international crude oil market and china’s commodity sectors: Evidence from time-frequency perspective of stochastic volatility[J]. Frontiers in Energy Research, 2020, 8(13): 45.

Tian Hongzhi, YAO Feng, Li Hui. Does China have international pricing power for crude oil? -- Based on the perspective of oil price independence and conductivity [J]. China Management Science, 2019,28(11):90-99.

LU F, YANG C, FANG L B. Do the crude oil futures of the shanghai International energy exchange improve asset allocation of Chinese petrochemical-related stocks? [J]. International Review of Financial Analysis, 2020, 71: 101537.

Zhang Dayong, Ji Qiang. Research on dynamic risk spillover of China's crude oil futures [J]. China Management Science,2018,26(11):42-49.

JIANG Y, JIANG C, NIE H, et al. The time-varying linkages between global oil market and China’s commodity sectors: Evidence from DCC-GJR-GARCH analyses[J]. Energy, 2019, 166: 577-586.

I hope the modification will do its best to meet your requirements.

Review Comment 4: Research methodology is not clear. it must be scientific and structured.

Reply to reviewer 1：Dear reviewer, thank you very much for pointing out my unclear problems. Based on your suggestions, I modified the model setting part, which mainly includes two parts, namely the wavelet transform part and the quantile Granger causality test part, and expounded the main steps of quantile Granger causality test, namely: First, the lag variable matrix is constructed, that is, the value of the past lag period of the two time series is taken as the independent variable.

Second, quantile regression of the constrained model is carried out, including only the lag variable of y, and the regression coefficient and residual are obtained.

Thirdly, quantile regression of the unrestricted model is carried out, including the lag variables of y and x, and the regression coefficient and residual are obtained.

Fourth, calculate the sum of residual squares of restricted and unrestricted models.

Fifth, calculate the T-test statistic, which is the difference of the sum of squares of the two model residuals divided by lag and then divided by the sum of squares of the unrestricted model residuals divided by the degrees of freedom.

Sixth, compare T-values, which are used to describe the degree of risk correlation between different futures varieties.

I hope the modification will do its best to meet your requirements.

Review Comment 5: discuss the key findings with literature in a separate section.

Reply to reviewer 1：Dear reviewer, thank you very much for pointing out the problem of "main findings of this paper compared with literatures in separate chapters". With reference to your suggestions, I selected five literatures with the highest similar length in section 6.1 of this paper and compared them from four aspects including research methods, research objects, research areas and research results, namely, the following aspects: In order to better illustrate the contribution of this paper, five literatures with the highest similarity to the research content of this paper are selected for comparison, which are Liu et al. (2016), Wang et al. (2017), Ma Yanran et al. (2020), Liu Yinglin et al. (2019) and Zhang Dayong et al. (2018). The research methods, research objects, research areas and research results are discussed as follows:

From the perspective of research methods, different methods and models were adopted in the five literatures to analyze the data, such as MGARCH model, DFA method, CVaR model, RSI method, MRCSB method, DCC model, ROM method and wavelet transform quantile Granger causality test. These methods and models have their advantages and disadvantages. For example, the MGARCH model can consider the volatility spillover and contagion effects among different varieties, but cannot distinguish the impact of positive and negative shocks. The DFA method can decompose the structural components of Chinese futures market, but can not capture the nonlinear relationship. CVaR model can measure the extreme risk level, but cannot reflect the direction of risk spillover. RSI method can measure the intensity of risk spillover, but can not reflect the persistence of risk spillover. The MRCSB method can identify the time and degree of the impact of international crude oil price fluctuations on China's commodity futures, but can not consider the multiple correlation structure. DCC model can capture dynamic conditional correlation, but can not distinguish upstream and downstream risk spillovers. ROM method can measure the dynamic risk spillover effect, but can not reflect the asymmetry of risk spillover effect. The wavelet transform quantile Granger causality test used in this paper can identify different frequency bands and different degrees of risk correlation.

From the perspective of research objects: the five papers all involve different varieties of China's futures market, such as crude oil, rubber, copper, aluminum, corn and soybeans, but their concerns are different. Liu et al. (2016) and Wang et al. (2017) both analyzed the volatility structure and dynamic characteristics of China's futures market as a whole. Ma Yanran et al. (2020), Liu Yinglin et al. (2019) and Zhang Dayong et al. (2018) respectively studied the risk spillover effects between domestic crude oil futures and other financial assets, international crude oil prices and China's commodity futures, and China's crude oil futures and other financial assets. This paper is based on wavelet transform and quantile Granger causality test to identify the risk correlation among precious metal futures market, crude oil futures market, grain futures market and common energy futures market.

From the study area: Liu et al. (2016) Wang et al. (2017) Ma Yanran et al. (2020) Liu Yinglin et al. (2019) Zhang Dayong et al. (2018) all studied the risk correlation among China's futures variety markets. This paper studies the risk correlation identification of the precious metal futures market, crude oil futures market, grain futures market and common energy futures market of 1 month and 6 month contract in the US

From the research results, the five papers have obtained some valuable results, such as the existence of significant volatility accumulation, leverage effect and persistence phenomenon in China's futures market; There is a common potential factor and a specific factor of each variety in China's futures market. There is a two-way extreme risk spillover effect between domestic crude oil futures and other financial assets, and it is affected by time period and market sentiment. The fluctuation of international crude oil prices has a significant positive impact on China's commodity futures, and there are differences in different varieties. There is a bidirectional dynamic risk spillover effect between China's crude oil futures and other financial assets, and it changes under different market sentiment. This paper draws the conclusion that there are different degrees and directions of risk correlation among precious metal futures market, crude oil futures market, grain futures market and common energy futures market, and it is stronger at high frequency band and high score.

I hope my modification can try my best to meet your requirements.

Review Comment 6: include implications and future scope of the study along with limitations.

Reply to reviewer 1：Dear reviewer, thank you very much for pointing out that "deficiencies, implications and future prospects of the research should be added to the end of the paper". Based on your suggestions, I add research implications, deficiencies and prospects of the research and other relevant contents at the end of the paper, as follows: This study provides enlightenments from the following aspects for relevant researchers:

First, different varieties, different frequency bands, different degrees and different directions of risk correlation in the futures market generally exist, which has important guiding significance for futures investors and regulators. Other researchers further explore the formation mechanism, influencing factors and dynamic changes of these risk associations in order to improve the risk identification and management ability of relevant entities in the futures market.

Second, this paper adopts the method of wavelet transform quantile Granger causality test, which is a novel method, which can effectively capture different frequency bands and different degrees of risk association. Other researchers can further compare the advantages and disadvantages of this method with other methods, as well as its applicability and robustness in different data and contexts.

Third, this paper deals with only four futures, namely gold, crude oil, soybeans and natural gas. Other researchers can further expand your research to include more futures varieties, such as copper, cotton, corn, etc., to test whether the results of this study hold true in other varieties and whether there is inter-variety heterogeneity.

Fourth, this paper only uses 1 month and 6 month futures contract data, other researchers can further consider longer or shorter maturities, such as 3 months, 9 months or 12 months, etc., to test whether the results of this study are consistent across different maturities and whether there is term structure effect.

Fifth, this paper only considers the impact of price changes on risk correlation, and other researchers can further consider other variables that may affect risk correlation, such as trading volume, open position, leverage ratio, liquidity, etc., in order to enrich the research content and conclusions on the topic of futures risk correlation.

Research deficiencies and prospects:

In this paper, the risk correlation among precious metal futures market, crude oil futures market, grain futures market and common energy futures market was identified only by constructing wavelet transformation-quantile Granger causality test, which provided a new perspective and method for the risk spillover effect of futures market and revealed the risk correlation of different varieties, different frequency bands, different degrees and different directions. However, there are some limitations in this paper, such as the applicability of wavelet transform quantile Granger causality test. Future studies can further expand the sample range, compare the effects of different methods, explore the risk transmission mechanism between different varieties, and introduce low-frequency macro variables with a high degree of correlation.

I hope the modification will do its best to meet your requirements.

Review Comment 7: proof editing is require to provide coherency in the entire manuscript.

Reply to reviewer 1：Dear reviewer, thank you for pointing out that "my paper needs polishing and editing". According to your suggestions, I have asked relevant institutions to revise the paper, eliminate grammatical and rhetorical errors as much as possible, and adjust the structure of the paper to meet your requirements as much as possible.

Review Comment 8: The author can perform Johnson Co-integration test to analyse the variables

Reply to reviewer 2：Dear reviewer, thank you for pointing out that "Johnson cointegration test should be added". With reference to your comments, I separately added the subsection of Johnson cointegration test. The results of cointegration test show that: Whether it is a 1-month futures contract or a 6-month futures contract, there is a certain degree of long-term relationship between gold futures, crude oil futures, soybean futures, natural gas futures and other variables.

I hope the modification will do its best to meet your requirements.

Review Comment 9: Conclusion need to be elaborate

Reply to reviewer 2：Dear reviewer, thank you for pointing out that "the conclusion section is too simple or vague, does not clearly summarize your findings and contributions, and does not indicate the limitations and future directions of your research." "In accordance with your reference comments, I re-summarize the conclusions in the conclusion section, the specific conclusions are as follows: First, whether it is a 1 month or 6 month contract, there is a strong two-way risk correlation between gold and natural gas.

Second, in the 1-month futures contract, there is an indirect two-way risk correlation between gold and crude oil, while in the 6-month futures, there is a strong one-way risk correlation between gold and crude oil, that is, the change of gold price will affect the change of crude oil price.

Third, in the 1-month futures contract, there is a strong direct bidirectional risk correlation between crude oil, soybeans and natural gas, while in the 6-month futures, there is a strong bidirectional risk correlation between crude oil and soybeans, but natural gas has a strong unidirectional risk correlation between crude oil and soybeans.

Fourth, there is a two-way risk association between gold and soybeans in both 1-month and 6-month futures contracts, but it is weaker at the 0.05 quantile and stronger at the 0.5 and 0.95 quantiles. This shows that the correlation between gold and soybean price movements increases with the degree of price volatility.

Fifth, at 0.95 quantile, there was a strong bidirectional or unidirectional risk association among all varieties.

Sixth, in the two futures varieties of 1-month contract and 6-month contract, there is a strong two-way risk correlation between crude oil and soybeans, that is, the price changes of crude oil and soybeans will affect each other.

Seventh, in the one-month contract, crude oil has a strong one-way risk correlation with natural gas, that is, the price change of crude oil will affect the price change of natural gas, but the reverse is not true.

Eighth, gold has a strong one-way risk correlation with crude oil in the six-month contract, that is, the price change of gold will affect the price change of crude oil, but the reverse is not true.

Ninth, in the 1-month contract and 6-month contract, soybean has a strong one-way risk correlation with gold, that is, the price change of soybean will affect the price change of gold, but the reverse is not true.

Tenth, at 0.5 and 0.95 quantiles, there is a strong two-way risk correlation between gold and natural gas, that is, the price changes of gold and natural gas will affect each other.

Eleventh, at 0.95 quantile, there is a strong bidirectional risk correlation between crude oil, soybeans and natural gas, that is, the price changes of crude oil, soybeans and natural gas will affect each other.

While adding the research results and contributions and the limitations and future directions of the research, the research results and contributions are as follows:

This study finds the above points, which are similar to the previous ones, such as the accumulation of volatility, leverage effect and persistence phenomenon in China's futures market; There is a common potential factor and a specific factor of each variety in China's futures market. There is a two-way extreme risk spillover effect between domestic crude oil futures and other financial assets, and it is affected by time period and market sentiment. The fluctuation of international crude oil prices has a significant positive impact on China's commodity futures, and there are differences in different varieties. There is a two-way dynamic risk spillover effect between China's crude oil futures and other financial assets, and the studies on the change under different market sentiment are different.

This paper identifies the risk correlation among precious metal futures market, crude oil futures market, grain futures market and common energy futures market by constructing wavelet transform and quantile Granger causality test, providing a new perspective and method for the risk spillover effect of futures market, and revealing the risk correlation of different varieties, different frequency bands, different degrees and different directions.

Limitations and future directions of the study are as follows:

There are also some limitations in this paper, such as the applicability of wavelet transform quantile Granger causality test method. Future studies can further expand the sample range, compare the effects of different methods, explore the risk transmission mechanism between different varieties, and introduce low-frequency macro variables with a high degree of correlation.

At the same time, it also points out in the conclusion: As can be seen from the above conclusions, the above conclusions show that this paper adopts wavelet transform and quantile Granger causality test to identify and measure the risk correlation of four major varieties (gold, crude oil, soybean and natural gas) in China's futures market, and reveals the differences and changes under different time frequencies and quantile levels. It verifies the research purpose proposed in the introduction of this paper and answers the questions about the risk correlation between different futures varieties and the change with time frequency and different quantile.

I hope the modification will do its best to meet your requirements.

Review Comment 10: Implications should be more specific

Reply to reviewer 2：Dear reviewer, thank you for your instruction that "the impact should be specific". I have added research enlightenment to the conclusion according to your suggestion, which is as follows:

This study provides implications for relevant researchers in the following aspects:

First, different varieties, different frequency bands, different degrees and different directions of risk correlation in the futures market generally exist, which has important guiding significance for futures investors and regulators. Other researchers further explore the formation mechanism, influencing factors and dynamic changes of these risk associations in order to improve the risk identification and management ability of relevant entities in the futures market.

Second, this paper adopts the method of wavelet transform quantile Granger causality test, which is a novel method, which can effectively capture different frequency bands and different degrees of risk association. Other researchers can further compare the advantages and disadvantages of this method with other methods, as well as its applicability and robustness in different data and contexts.

Third, this paper deals with only four futures, namely gold, crude oil, soybeans and natural gas. Other researchers can further expand your research to include more futures varieties, such as copper, cotton, corn, etc., to test whether the results of this study hold true in other varieties and whether there is inter-variety heterogeneity.

Fourth, this paper only uses 1 month and 6 month futures contract data, other researchers can further consider longer or shorter maturities, such as 3 months, 9 months or 12 months, etc., to test whether the results of this study are consistent across different maturities and whether there is term structure effect.

Fifth, this paper only considers the impact of price changes on risk correlation, and other researchers can further consider other variables that may affect risk correlation, such as trading volume, open position, leverage ratio, liquidity, etc., in order to enrich the research content and conclusions on the topic of futures risk correlation.

I hope the modification will do its best to meet your requirements.

Review Comment 11: Gramatical errors were found

Reply to reviewer 2：Dear reviewer, thank you for pointing out that "there are grammar errors". With reference to your revision suggestions, I have asked relevant institutions to revise it to reduce grammar errors, hoping that I can try my best to meet your requirements.

Review Comment 12: Latest Literature review can be added

Reply to reviewer 2：Dear reviewer, thank you very much for pointing out that "recent relevant literatures should be added". I have added 11 literatures on related topics in reference to your suggestion, and the specific literatures are as follows:

Ma Yanran, Hu Min, Zhang Dayong, et al. Extreme risk spillover of domestic crude oil futures and other financial assets [J]. Research of Environmental Economics,20,5(3):115-132.

YANG Y Y, MA Y R, HU M, et al. Extreme risk spillover between chinese and global crude oil futures[J]. Finance Research Letters, 2021, 40: 101743.

Liu Yinglin, LIU Yonghui, JU Zhuo. The impact of international crude oil price fluctuation on China's commodity futures: An analysis based on multiple correlation structure breakpoints [J]. Chinese Journal of Management Science,2019,27(2):31-40.

KANG S H, MCIVER R, YOON S M. Dynamic spillover effects among crude oil, precious metal, and agricultural commodity futures markets[J]. Energy Economics, 2017, 62: 19-32.

WU B B. The dynamics of oil on China’s commodity sectors: What can we learn from a quantile perspective? [J]. Journal of Commodity Markets, 2021, 23: 100158.

MENG J, NIE H, MO B, et al. Risk spillover effects from global crude oil market to china’s commodity sectors[J]. Energy, 2020, 202: 117208.

AHMED A D, HUO R. Volatility transmissions across international oil market, commodity futures and stock markets: Empirical evidence from China[J]. Energy Economics, 2021, 93: 104741.

LI Z H, SU Y Y. Dynamic spillovers between international crude oil market and china’s commodity sectors: Evidence from time-frequency perspective of stochastic volatility[J]. Frontiers in Energy Research, 2020, 8(13): 45.

Tian Hongzhi, YAO Feng, Li Hui. Does China have international pricing power for crude oil? -- Based on the perspective of oil price independence and conductivity [J]. China Management Science, 2019,28(11):90-99.

LU F, YANG C, FANG L B. Do the crude oil futures of the shanghai International energy exchange improve asset allocation of Chinese petrochemical-related stocks? [J]. International Review of Financial Analysis, 2020, 71: 101537.

Zhang Dayong, Ji Qiang. Research on dynamic risk spillover of China's crude oil futures [J]. China Management Science,2018,26(11):42-49.

JIANG Y, JIANG C, NIE H, et al. The time-varying linkages between global oil market and China’s commodity sectors: Evidence from DCC-GJR-GARCH analyses[J]. Energy, 2019, 166: 577-586.

I hope the modification will do its best to meet your requirements.

---

## [Decision Letter · Decision Letter 1]

26 Oct 2023

Risk correlation identification of futures market based on wavelet transform and quantile Granger causality test

PONE-D-23-30906R1

Dear Dr. wu,

We’re pleased to inform you that your manuscript has been judged scientifically suitable for publication and will be formally accepted for publication once it meets all outstanding technical requirements.

Kind regards,

Chinnadurai Kathiravan

Academic Editor

PLOS ONE

Additional Editor Comments (optional):

Reviewers' comments:

Reviewer's Responses to Questions

**Comments to the Author**

1. If the authors have adequately addressed your comments raised in a previous round of review and you feel that this manuscript is now acceptable for publication, you may indicate that here to bypass the “Comments to the Author” section, enter your conflict of interest statement in the “Confidential to Editor” section, and submit your "Accept" recommendation.

Reviewer #1: All comments have been addressed

Reviewer #2: All comments have been addressed

2. Is the manuscript technically sound, and do the data support the conclusions?

Reviewer #1: Yes

Reviewer #2: Yes

3. Has the statistical analysis been performed appropriately and rigorously? 

Reviewer #1: Yes

Reviewer #2: Yes

4. Have the authors made all data underlying the findings in their manuscript fully available?

Reviewer #1: Yes

Reviewer #2: Yes

5. Is the manuscript presented in an intelligible fashion and written in standard English?

Reviewer #1: Yes

Reviewer #2: Yes

6. Review Comments to the Author

Reviewer #1: The topic is relevant and authors have address all the comments. I recommend this paper for publication.

Reviewer #2: The author incorporated suggested corrections including literature review, research questions, research Gap, Methodoogy Analysis and Elaborated the conclusion.

7. PLOS authors have the option to publish the peer review history of their article (what does this mean?). If published, this will include your full peer review and any attached files.

Reviewer #1: No

Reviewer #2: **Yes: **Rajesh Mamilla

---

## [Editor Report · Acceptance letter]

9 Nov 2023

PONE-D-23-30906R1 

Risk correlation identification of futures market based on wavelet transform and quantile Granger causality test 

Dear Dr. Wu:

I'm pleased to inform you that your manuscript has been deemed suitable for publication in PLOS ONE. Congratulations! Your manuscript is now with our production department. 

Kind regards, 

on behalf of

Dr. Chinnadurai Kathiravan 

Academic Editor

PLOS ONE